# The pan-cancer lncRNA PLANE regulates an alternative splicing program to promote cancer pathogenesis

Liu Teng[1,10], Yu Chen Feng[2,10], Su Tang Guo[3], Pei Lin Wang[1], Teng Fei Qi[1], Yi Meng Yue[1], Shi Xing Wang[1], Sheng Nan Zhang[1], Cai Xia Tang[1], Ting La[4], Yuan Yuan Zhang[4], Xiao Hong Zhao[4], Jin Nan Gao[5], Li Yuan Wei[5], Didi Zhang[6], Jenny Y. Wang[7], Yujie Shi[8], Xiao Ying Liu[1], Jin Ming Li[1], Huixia Cao[9], Tao Liu[1,7], Rick F. Thorne[1,4], Lei Jin[1,2✉], Feng-Min Shao[9✉] & Xu Dong Zhang[1,4✉]

Genomic amplification of the distal portion of chromosome 3q, which encodes a number of oncogenic proteins, is one of the most frequent chromosomal abnormalities in malignancy. Here we functionally characterise a non-protein product of the 3q region, the long noncoding RNA (lncRNA) PLANE, which is upregulated in diverse cancer types through copy number gain as well as E2F1-mediated transcriptional activation. PLANE forms an RNA-RNA duplex with the nuclear receptor co-repressor 2 (NCOR2) pre-mRNA at intron 45, binds to heterogeneous ribonucleoprotein M (hnRNPM) and facilitates the association of hnRNPM with the intron, thus leading to repression of the alternative splicing (AS) event generating NCOR2-202, a major protein-coding NCOR2 AS variant. This is, at least in part, responsible for PLANE-mediated promotion of cancer cell proliferation and tumorigenicity. These results uncover the function and regulation of PLANE and suggest that PLANE may constitute a therapeutic target in the pan-cancer context.

[1] Translational Research Institute, Henan Provincial People's Hospital and People's Hospital of Zhengzhou University, Academy of Medical Science, Zhengzhou University, Henan, China. [2] School of Medicine and Public Health, The University of Newcastle, Callaghan, NSW, Australia. [3] Department of Molecular Biology, Shanxi Cancer Hospital and Institute, Shanxi, China. [4] School of Biomedical Sciences and Pharmacy, The University of Newcastle, Callaghan, NSW, Australia. [5] Department of Breast Surgery, Shanxi Bethune Hospital, Shanxi, China. [6] Orthopaedics Department, John Hunter Hospital, Hunter New England Health, New Lambton, NSW, Australia. [7] Children's Cancer Institute Australia for Medical Research, University of New South Wales, Sydney, NSW, Australia. [8] Department of Pathology, Henan Provincial People's Hospital, Zhengzhou University People's Hospital, Henan, China. [9] Department of Nephrology, Henan Provincial People's Hospital, Zhengzhou University People's Hospital, Henan, China. [10] These authors contributed equally: Liu Teng, Yu Chen Feng. ✉email: Lei.Jin@newcastle.edu.au; fengminshao@126.com; Xu.Zhang@newcastle.edu.au

Alternative splicing (AS) of precursor mRNAs (pre-mRNAs) is a fundamental mechanism that allows for the generation of diverse mature transcripts from a single gene thus amplifying the gene-coding capacity and increasing the functional diversity[1–3]. Over 95% of human multiexon genes undergo AS that is tightly controlled by the interaction of trans-acting proteins referred to as splicing factors with cis-acting nucleotide sequences[1–3]. Splicing factors encompass members of the serine-arginine (SR) protein family and heterogeneous ribonucleoproteins (hnRNPs) that promote or repress specific splicing events through interacting with exonic or intronic regulatory sequences classified as enhancers or silencers[1,4]. Aberrant AS events are involved in the pathogenesis of many diseases including cancer through deregulating essential cellular processes such as cell survival and proliferation[1,4–6].

Nuclear receptor co-repressor 2 (NCOR2), also known as silencing mediator of retinoid and thyroid hormone receptors (SMRT) or T$_3$ receptor-associating cofactor 1 (TRAC-1), acts as a central organising platform for assembling functional complexes which repress the transactivation of target genes. The NCOR2 N-terminal repression domains recruit other transcriptional co-repressors such as histone deacetylases (HDACs) while its C-terminal interaction domains interact with nuclear receptors such as the thyroid hormone receptor and retinoic acid receptor[7–9]. Moreover, NCOR2-mediated repression also targets genes activated by other transcription factors such as AP-1 and NF-κB[9–11]. With its repression domains retained, NCOR2 exhibits varying affinities for different transcription factors through AS at its C-terminus[9,12,13]. Of note, cancer cells often display altered expression of NCOR2, implicating a role of its deregulation in cancer pathogenesis[14–18]. For example, NCOR2 is downregulated in multiple myeloma and its low expression is associated with the development of non-Hodgkin's lymphoma and poor prognosis of lung adenocarcinoma (LUAD) patients[14–17]. In contrast, high NCOR2 expression is linked to earlier recurrence of breast carcinoma (BRCA)[18].

There is increasing appreciation of the role of long noncoding RNAs (lncRNAs) in cancer development and progression[1,19–22]. In particular, a growing number of lncRNAs have been linked to the deregulation of AS in cancer cells[23–28]. LncRNAs regulate AS primarily through binding to splicing factors, associating with pre-mRNAs and impinging on chromatin remodelling[24]. For instance, the lncRNA MALAT1 regulates alternative splicing of a set of pre-mRNAs through modulating SR splicing factor phosphorylation and sub-nuclear localization, and is thus involved in the development, progression and treatment resistance of many types of cancers[26], whereas the lncRNA LNIC01133 interacts with the SR splicing factor SRSF6 (SRp55) resulting in inhibition of epithelial-mesenchymal transition (EMT) and metastasis[27]. Moreover, the lncRNA SAF binds to the *Fas* pre-mRNA and recruits splicing factor 45 (SPF45) leading to generation of a Fas AS variant that protects cancer cells from Fas-induced cell death[28].

Here we present evidence that the lncRNA PLANE forms an RNA–RNA duplex with the NCOR2 pre-mRNA and recruits hnRNPM, thus facilitating hnRNPM-mediated repression of the AS event generating NCOR2-202, a major protein-coding NCOR2 transcript variant. The resulting downregulation of NCOR2 at the protein level contributes to the increased proliferation and tumorigenicity of cancer cells. Moreover, we show that PLANE is frequently upregulated in diverse cancer types through genomic amplification and E2F1-mediated transcriptional activation, with practical implications of interference with PLANE as potential treatment approach in the pan-cancer context.

## Results

**Genomic amplification and transcriptional activation by E2F1 drive PLANE upregulation in diverse cancer types.** Through interrogating the lncRNA expression data in the Cancer Genome Atlas (TCGA)[29], we identified a panel of 18 pan-cancer upregulated lncRNAs that were increased in expression in at least 19 of 20 cancer types in relation to corresponding normal tissues (Supplementary Fig. 1a). Among them was melanotransferrin (MELTF, also known as MFI2) antisense RNA1 (MELTF-AS1 or MFI2-AS1) that is encoded by a gene located to the distal portion of chromosome 3q (3q29) (Supplementary Fig. 1a–c), whose amplification is one of the most prevalent chromosomal abnormalities observed in various cancer types[30–34]. Indeed, *MELTF-AS1* was the most frequently amplified gene among those that encode the pan-cancer upregulated lncRNAs (Supplementary Fig. 1d). We therefore sought to investigate the potential role of MELTF-AS1 in cancer pathogenesis. Of the five annotated MELTF-AS1 isoforms (Vega Genome Browser), the longest isoform was markedly more abundant than others in multiple cancer cell lines, including A549 and H1299 LUAD, NCI-H226 lung squamous cell carcinoma (LUSC), HCT116 colon adenocarcinoma (COAD), MCF-7 BRCA and Eca109 esophageal squamous cell carcinoma (ESCC), as shown in RT-PCR analysis with isoform targeting primers including those recognise a region spanning across exon 2 and 3 (Supplementary Fig. 1e, f). We hereafter focused on this isoform and renamed it PLANE (Pan-cancer LncRNA Activating NCOR2 responsive to E2F1) given its functional relationship with NCOR2 and transcriptional responsiveness to E2F1 (see below). PLANE consists of four exons (E1–E4), with minimum free energy modelling predicting a broadly symmetrical structure with E1 and E3 constituting each pole, whereas E2 and E4 contributing to both poles of the molecule (Supplementary Fig. 1g).

We confirmed the cancer-associated upregulation of PLANE in cohorts of formalin-fixed paraffin-embedded (FFPE) LUSC, LUAD and COAD and freshly isolated ESCC and BRCA samples compared to adjacent normal tissues (Fig. 1a and Supplementary Fig. 2a). Noticeably, despite the common increase in cancer tissues, PLANE levels did not differ among tumours of different stages (Supplementary Fig. 2b and Supplementary Tables 1–3). Likewise, there were no significant differences in PLANE expression between LUSC, LUAD and COAD of different groups stratified by tumour grade and patient gender as well as their median age at diagnosis (Supplementary Tables 1–3). Moreover, no significant changes were found in PLANE expression levels between COAD and colon adenomas (pre-neoplastic colon lesions), whereas PLANE expression was increased in colon adenomas compared with normal colon epithelia (Supplementary Fig. 2c). Collectively, these results suggest that PLANE upregulation is an early event during tumorigenesis. Furthermore, high PLANE expression was associated with poorer overall patient survival (OS) in diverse cancer types (Fig. 1b and Supplementary Fig. 2d), implicating its broad involvement in cancer development and progression.

Consistent with the contribution of genomic amplification to the upregulation of PLANE in some cancer tissues (Supplementary Fig. 1d), qPCR analysis of genomic DNA demonstrated *PLANE* copy number gains in ~36% of LUSC (8 of 22) and ~4% of LUAD (1 of 24) (Fig. 1c). Increases in *PLANE* copy numbers were also evident in 2 of 7 cancer cell lines compared with the CCC-HIE-2 normal human intestinal epithelial cell line (Fig. 1d). Nonetheless, similar to the pan-cancer upregulation of PLANE in tissue samples, all the cancer cell lines examined expressed higher levels of PLANE than the CCC-HSF-1 normal skin fibroblast cell line irrespective of their amplification status (Supplementary Fig. 2e), suggesting that

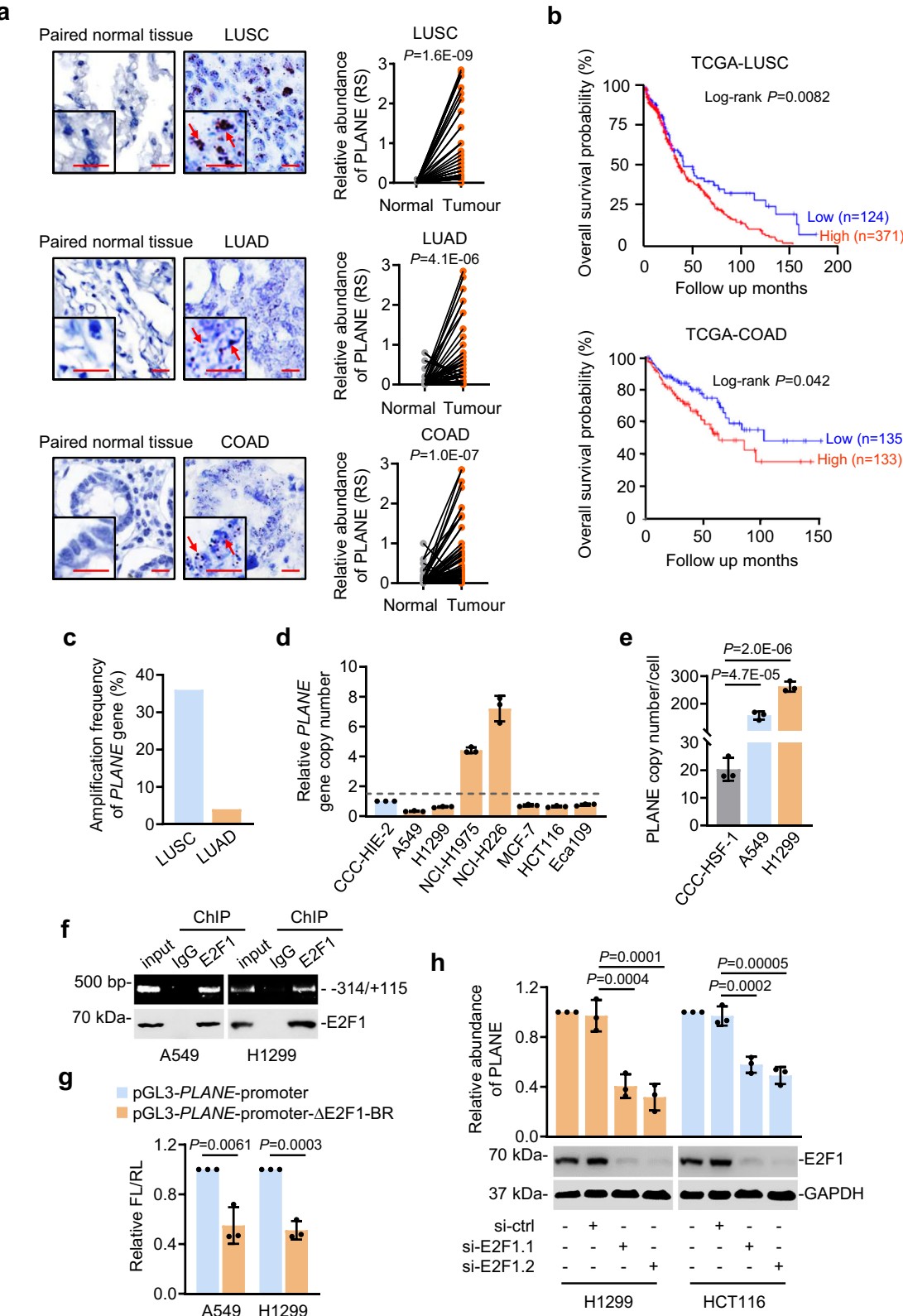

additional causal mechanisms such as transcriptional regulation are involved in the upregulation of PLANE in cancer cells. In support, absolute quantitation showed that there were ~157 and ~262 PLANE molecules per A549 and H1299 cell, respectively, which did not have copy number gains in the *PLANE* gene, compared with ~20 PLANE molecules per CCC-HSF-1 cell (Fig. 1e).

To gain insights into transcriptional mechanisms involved in PLANE upregulation in cancer cells, we analysed its promoter for transcription factor binding sites using bioinformatics. This predicted multiple E2F1-binding motifs located to the −314/−14 region of the proximal promoter of the *PLANE* gene (Supplementary Fig. 3a)[35]. Indeed, this region (E2F1-BR) was

**Fig. 1 Genomic amplification and transcriptional activation by E2F1 drive PLANE expression that is upregulated in diverse cancer types. a** Representative microphotographs and quantitation of in situ hybridization (ISH) analysis of PLANE expression in formalin-fixed paraffin-embedded (FFPE) LUSC, LUAD and COAD tissues ($n = 75$, 87 and 79 biologically independent samples, respectively) compared with corresponding paired adjacent normal tissues. Scale bar, 5 μm. RS: reactive score. Two-tailed Student's $t$-test. **b** Kaplan–Meier analysis of the probability of overall survival of LUSC ($n = 495$) and COAD ($n = 268$) patients derived from the TCGA datasets. The log-rank test. **c** qPCR analysis of genomic DNA from LUSC ($n = 22$) and LUAD ($n = 24$) tissues and corresponding paired adjacent normal tissues. A ≥1.5-fold increase in the copy number is considered genomic amplification. **d** qPCR analysis of genomic DNA from the indicated cancer cell lines and the normal human intestinal epithelial cell line CCC-HIE-2. The copy number of PLANE in the CCC-HIE-2 cell line was arbitrarily designated as 1. A ≥1.5-fold increase in the copy number is considered genomic amplification. Data are mean ± s.d.; $n = 3$ independent experiments. **e** Absolute quantitation of PLANE in A549 and H1299 cancer cells and normal human CCC-HSF-1 fibroblasts using qPCR. Data are mean ± s.d.; $n = 3$ independent experiments, one-way ANOVA followed by Tukey's multiple comparisons test. **f** Chromatin immunoprecipitation (ChIP) analysis of the association between endogenous E2F1 and the E2F1-binding motifs at the promoter of PLANE in A549 and H1299 cells. Data are representatives of three independent experiments. **g** The transcriptional activity of a PLANE reporter construct was reduced by deletion of the E2F1-binding region (E2F1-BR) at the promoter of PLANE in A549 and H1299 cells. Data are mean ± s.d.; $n = 3$ independent experiments, two-tailed Student's $t$-test. FL: Firefly luciferase activity; RL: Renilla luciferase activity. **h** E2F1 silencing downregulated PLANE expression in H1299 and HCT116 cells. Data are representatives or mean ± s.d.; $n = 3$ independent experiments, one-way ANOVA followed by Tukey's multiple comparisons test. Source data are provided as a Source data file.

co-precipitated with endogenous E2F1 and was required for transcriptional upregulation of PLANE as the transcriptional activity of a PLANE reporter construct was inhibited when the E2F1-BR was deleted (Fig. 1f, g). Moreover, co-transfection of E2F1 selectively enhanced the transcriptional activity of the PLANE reporter whereas knockdown of E2F1 diminished reporter activity (Supplementary Fig. 3b, c), supporting the notion that PLANE is transcriptionally activated by E2F1 through the identified E2F1-BR. In accordance, knockdown of E2F1 reduced the endogenous PLANE levels, whereas overexpression of E2F1 caused PLANE upregulation (Fig. 1h and Supplementary Fig. 3d). There was a trend that PLANE levels were positively associated with the levels of E2F1 in diverse cancer types in TCGA datasets (Supplementary Fig. 3e). Furthermore, qPCR analysis of cohorts of freshly isolated ESCC and LUAD samples showed that the levels of PLANE were indeed correlated with E2F1 expression levels (Supplementary Fig. 3f). Collectively, these results demonstrate that E2F1 along with genomic amplification are responsible for the upregulation of PLANE in cancer cells.

The gene encoding PLANE is divergently located opposite to the protein-coding gene MELTF (Supplementary Fig. 4a). Notably, high MELTF mRNA expression is also negatively associated with survival of patients with diverse types of cancers in the TCGA datasets (Supplementary Fig. 4b). Nevertheless, knockdown of PLANE did not impinge upon MELTF mRNA expression (Supplementary Fig. 4c), and similarly, knockdown of MELTF did not affect PLANE expression levels (Supplementary Fig. 4d). Thus, there is no regulatory interaction between PLANE and its neighbouring gene MELTF and the impact of PLANE on patient survival is not associated with the relationship between MELTF mRNA expression and patient prognosis.

**PLANE promotes cancer cell proliferation and tumorigenicity.** We examined the biological significance of PLANE upregulation in cancer cells. SiRNA knockdown of PLANE inhibited cell proliferation and reduced clonogenicity in diverse cancer cell lines (Fig. 2a–c), which was associated with G0/G1 cell cycle arrest (Supplementary Fig. 5a). Conversely, overexpression of PLANE increased, albeit moderately, proliferation in A549 and H1299 cells (Supplementary Fig. 5b). Gene set enrichment analysis (GSEA) of the RNA-sequencing (RNA-seq) data from A549 cells revealed that knockdown of PLANE caused downregulation of numerous genes of signalling pathways involved in cell cycle progression, including the E2F1, G2/M checkpoint and mitotic spindle assembly pathways (Supplementary Fig. 5c). Thus, PLANE expression promotes the integral proliferative machinery of cancer cells.

To facilitate further investigations, we established A549 and H1299 sublines (A549.shPLANE and H1299.shPLANE) with conditional knockdown of PLANE in response to doxycycline (Dox) (Fig. 2d). Induced PLANE knockdown similarly triggered reductions in cell proliferation and clonogenicity and induced anchorage-independent growth in A549.shPLANE and H1299. shPLANE sublines (Fig. 2e–g and Supplementary Fig. 5d). Moreover, Dox treatment of nu/nu mice retarded the growth of A549.shPLANE.1 xenografts (Fig. 2h, i and Supplementary Fig. 5e, f). Cessation of Dox treatment restored the expression of PLANE and recovered, at least in part, the clonogenic potential in vitro and tumour xenograft growth in mice (Fig. 2f–i and Supplementary Fig. 5e, f), further consolidating the role of PLANE in tumorigenicity.

**PLANE regulates NCOR2 pre-mRNA AS.** To dissect the mechanisms whereby PLANE promotes cancer cell proliferation, we compared the transcriptomes of A549 cells with and without PLANE knockdown using two different siRNAs. Our initial short-read (150 bp) RNA-seq data showed that the NCOR2 mRNA variant, NCOR2-202 (ENST00000397355.1; grch37. ensembl.org), was the most highly upregulated transcript among those (44 transcripts) that were commonly increased after PLANE knockdown with two different siRNAs (Fig. 3a and Supplementary Data 1). Thereafter, long-read RNA-seq to obtain full-length transcript data confirmed that NCOR2-202 was upregulated after PLANE knockdown (Fig. 3b). In contrast, neither short- nor long-read RNA-seq identified significant changes in the levels of NCOR2-001 and NCOR2-005, which along with NCOR2-202, give rise to the three major NCOR2 protein isoforms, NCOR2-3, -1 and -2, respectively (Fig. 3a, b, Supplementary Fig. 6 and Supplementary Data 1)[36]. Intriguingly, knockdown or overexpression of PLANE did not alter NCOR2 pre-mRNA expression (Supplementary Fig. 7a, b). Moreover, knockdown of PLANE did not affect the enrichment of the transcriptional activation mark H3K4me3 and the transcriptional repression mark H3K27me3 at the NCOR2 promoter (Supplementary Fig. 7c)[37]. These results suggest that PLANE may selectively inhibit NCOR2-202 expression through post-transcriptional regulation.

The NCOR2-202 mRNA variant differs from NCOR2-001 and NCOR2-005 in its generation through an alternative 5′ splice site within intron 45 (Supplementary Fig. 8a). Noticeably, the NCOR2-017 mRNA variant is also spliced out from this site (Supplementary Fig. 8a). On the other hand, similar to NCOR2-001 and NCOR2-005, a number of additional mRNA variants, including NCOR2-002, NCOR2-018, NCOR2-201 and

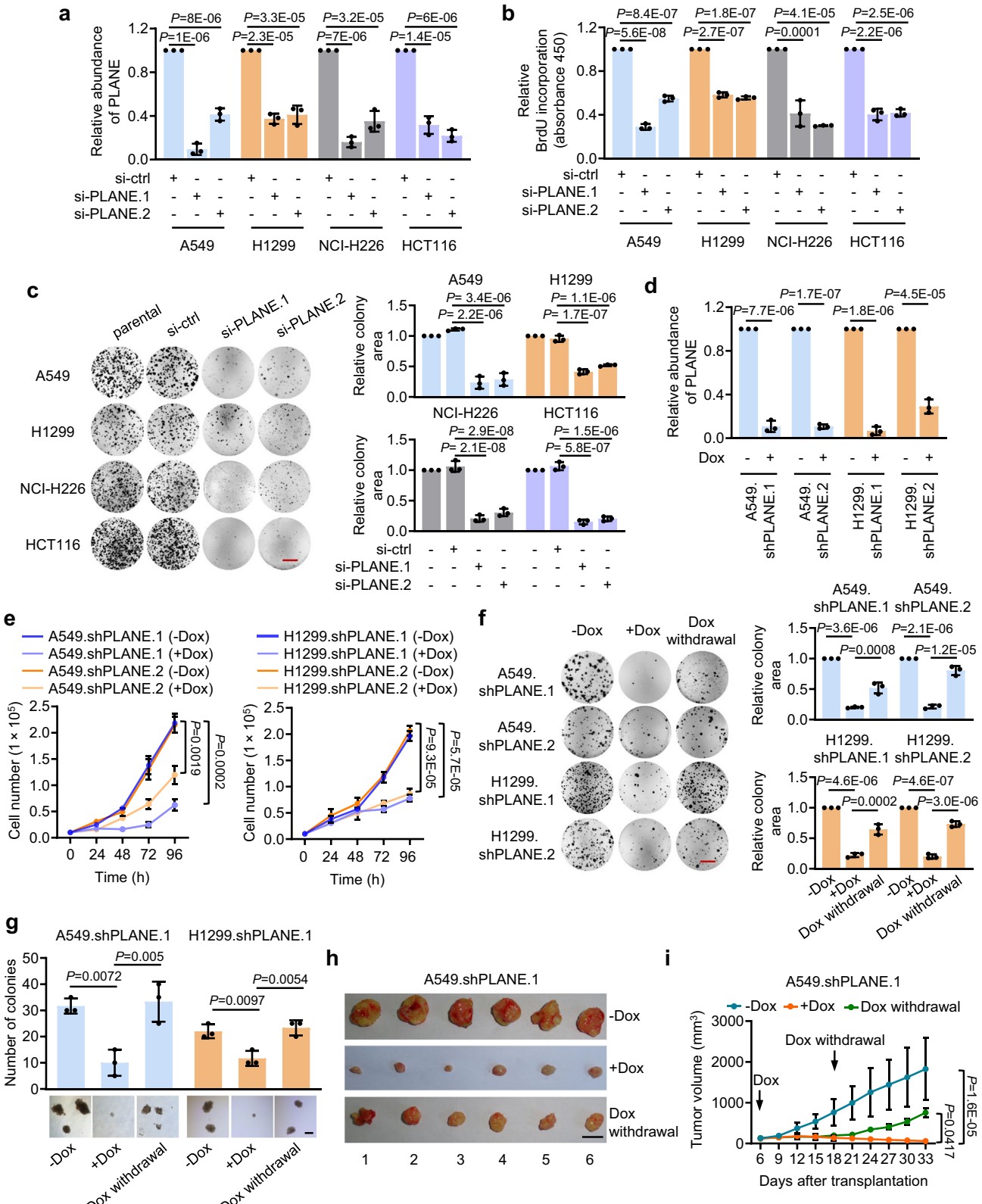

NCOR2-203 involve splicing at the classic boundary of exon and intron 45. For simplicity, we hereafter collectively refer to these mRNAs as NCOR2-001/005-like mRNA variants (Supplementary Fig. 8a). Due to sequence overlaps it was not feasible to specifically detect NCOR2-202 using qPCR to verify directly its increase caused by PLANE knockdown (Supplementary Fig. 6). However, by use of primers recognising the 1−138 fragment of

intron 45 that is not contained by NCOR2-202, we could collectively detect the NCOR2-001/005-like mRNA variants using qPCR (Supplementary Fig. 8a). The results exhibited a decrease, albeit moderately, in cells with PLANE knockdown (Fig. 3c). In contrast, qPCR analysis using primers recognising a common region present in NCOR2-202 and the NCOR2-001/005-like mRNA variants except for NCOR2-018 revealed an increase in

**Fig. 2 PLANE promotes cancer cell proliferation and tumorigenicity. a–c** SiRNA knockdown of PLANE (**a**) inhibited 5-bromo-2′-deoxyuridine (BrdU) incorporation (**b**) and clonogenicity (**c**) in multiple cancer cell lines. Relative clonogenicity was quantitated using ImageJ-plugin 'ColonyArea'. Data are mean ± s.d. or representatives; $n = 3$ independent experiments, one-way ANOVA followed by Tukey's multiple comparison test. Scale bar, 1 cm. **d** Induced knockdown of PLANE by the addition of doxycycline (Dox, 500 nM) in A549.shPLANE and H1299.shPLANE cells. Data are mean ± s.d.; $n = 3$ independent experiments, two-tailed Student's $t$-test. **e** Dox (500 nM)-induced knockdown of PLANE inhibited A549.shPLANE and H1299.shPLANE cell proliferation as shown by decelerated cell number increases. Data are mean ± s.d.; $n = 3$ independent experiments, two-tailed Student's $t$-test. **f** Induced knockdown of PLANE inhibited A549.shPLANE and H1299.shPLANE cell clonogenicity, which was partially reversed by cession of Dox (500 nM) treatment. Relative clonogenicity of cells was quantitated using ImageJ-plugin 'ColonyArea'. Data are representatives or mean ± s.d.; $n = 3$ independent experiments, one-way ANOVA followed by Tukey's multiple comparisons test. Scale bar, 1 cm. **g** Representative microscopic photographs of anchorage-independent growth of A549.shPLANE and H1299.shPLANE cells with or without treatment with Dox and cessation of Dox treatment. Quantification of anchorage-independent growth of the cells is also shown. Scale bar, 0.5 mm. Data are representatives or mean ± s.d.; $n = 3$ independent experiments, one-way ANOVA followed by Tukey's multiple comparisons test. **h, i** Representative photographs (**h**) and growth curves (**i**) of A549.shPLANE xenografts in nu/nu mice with or without treatment with Dox (2 mg/ml supplemented with 10 mg/ml sucrose in drinking water) and cessation of Dox treatment. Data are representatives or mean ± s.d.; $n = 6$ mice per group, one-way ANOVA followed by Tukey's multiple comparison test. DOX: 2 mg/ml supplemented with 10 mg/ml sucrose in drinking water. Source data are provided as a Source data file.

expression caused by knockdown of PLANE (Fig. 3d and Supplementary Fig. 6). Importantly, qPCR analysis using primers recognising only NCOR2-018 and NCOR2-203 displayed reduced expression in PLANE knockdown cells (Supplementary Fig. 8b). Moreover, analysis using primers specifically directed to NCOR2-017 showed similarly low levels of expression in cells with and without knockdown of PLANE (Supplementary Fig. 8c). Taken together, these results indicate that PLANE indeed selectively inhibits the expression of the NCOR2-202 mRNA variant (Supplementary Fig. 8d). Nonetheless, there was no significant relationship between PLANE and NCOR2-202 expression levels in freshly isolated ESCC and LUAD samples as shown with semi-quantitative RT-PCR (Supplementary Fig. 8e), implying that additional mechanisms are likely involved in regulating NCOR2-202 mRNA expression.

We further interrogated the RNA-seq data to verify that regulation of NCOR2-202 expression by PLANE is caused by alterations in AS of the NCOR2 pre-mRNA. Quantitative visualisation of AS frequency within the *NCOR-2* gene using integrated genome viewer (IGV)-Sashimi plot revealed that the frequency of the AS event generating NCOR2-202 and NCOR2-017 was increased, whereas the frequency of the AS event giving rise to the NCOR2-001/005-like mRNA variants was reduced in cells with PLANE knockdown (Fig. 3e and Supplementary Data 2). To substantiate these data, we carried out semi-quantitative RT-PCR analysis using primers flanking the splice site that generates NCOR2-202 and NCOR2-017 (Supplementary Fig. 8a). Detection of the splicing event generating the NCOR2-001/005-like mRNA variants at the junction of exon/intron 45 was included as a control. Instructively, PLANE knockdown resulted in an increase in the NCOR2-202- and NCOR2-017-generating AS event and a decrease in the event giving rise to the NCOR2-001/005-like mRNA variants (Fig. 3f, g). In contrast, overexpression PLANE decreased the NCOR2-202- and NCOR2-017-generating AS event with accompanying moderate increases in AS event giving rise to the NCOR2-001/005-like mRNA variants (Fig. 3h, i). Thus, PLANE inhibits NCOR2-202 expression through repressing its generation by AS (Supplementary Fig. 8d).

We next examined whether PLANE-mediated regulation of the NCOR2-202-generating AS event results in changes in NCOR2 protein expression. Immunoblotting with NCOR2 antibodies against residues near its N-terminus that are conserved in NCOR2 protein isoforms 1–3 demonstrated that knockdown of PLANE indeed caused an increase in NCOR2 protein levels (Figs. 2d, 3d and Supplementary Fig. 6), whereas overexpression of PLANE led to downregulation of NCOR2 protein levels (Supplementary Fig. 9a), implicating that regulation of NCOR2-

202 AS variant expression is indeed associated with changes in NCOR2 protein expression. We also endeavoured to use siRNAs to selectively target the 1–138 fragment within intron 45 present in NCOR2-001 and NCOR2-005 but not NCOR2-202, an approach that would further interrogate the contribution of the increase in NCOR2-202 to the upregulation of NCOR2 protein expression caused by PLANE knockdown (Supplementary Table 4). However, we did not achieve meaningful knockdown of NCOR2-001 and/or NCOR2-005 expression after numerous attempts with two different siRNAs (Supplementary Fig. 9b). Nevertheless, among the three major protein-producing NCOR2 mRNA variants, NCOR2-001 and NCOR2-005 were present at relatively high levels, whereas NCOR2-202 levels, similar to the levels of the NCOR2 protein, were low in cells without knockdown of PLANE (Fig. 3d, f, g), suggesting that NCOR2 protein expression is not closely related the expression of the NCOR2-001 and NCOR2-005 mRNA variants, presumably due to mechanisms regulating their further processing, translation and/or posttranslational stabilisation[38,39]. On the other hand, while the NCOR2-001 and NCOR2-005 levels remained unchanged, the levels of NCOR2-202 was apparently increased, concurring with the upregulation of the NCOR2 protein in cells with PLANE knocked down (Fig. 3a, b, d, f, g and Supplementary Data 1). Collectively, these results indicate that PLANE-mediated inhibition of NCOR2 protein expression is associated with repression of the AS event producing NCOR2-202.

We also considered if PLANE affected the regulation of NCOR2 mRNA nuclear export which would also influence NCOR2 protein expression. A qPCR strategy using primers detecting a common region of the major protein-coding NCOR2 mRNA variants NCOR2-001, NCOR2-005 and NCOR2-202 as well as NCOR2-015, NCOR2-017, NCOR-018, NCOR2-022, NCOR2-201 and NCOR2-203 was used against subcellular fractions in A549 cells with or without knockdown of PLANE. The results showed that PLANE knockdown did not cause any significant changes in the proportions of these NCOR2 AS variants distributing to the nuclear and cytoplasmic fractions (Supplementary Fig. 9c). Thus, PLANE-mediated regulation of NCOR2 expression is not associated with altered nuclear export of these NCOR2 mRNA variants.

To investigate the biological impact of PLANE regulation of AS production of NCOR2-202, we tested the effect of siRNAs targeting common regions of NCOR2-001, NCOR2-005 and NCOR2-202 on inhibition of cell proliferation caused by PLANE knockdown (Supplementary Fig. 6), which conceivably reflected the consequence of inhibition NCOR2-202 expression, as NCOR2 protein upregulation in PLANE knockdown cells was closely associated with the increase in NCOR2-202 (Fig. 3a–g and

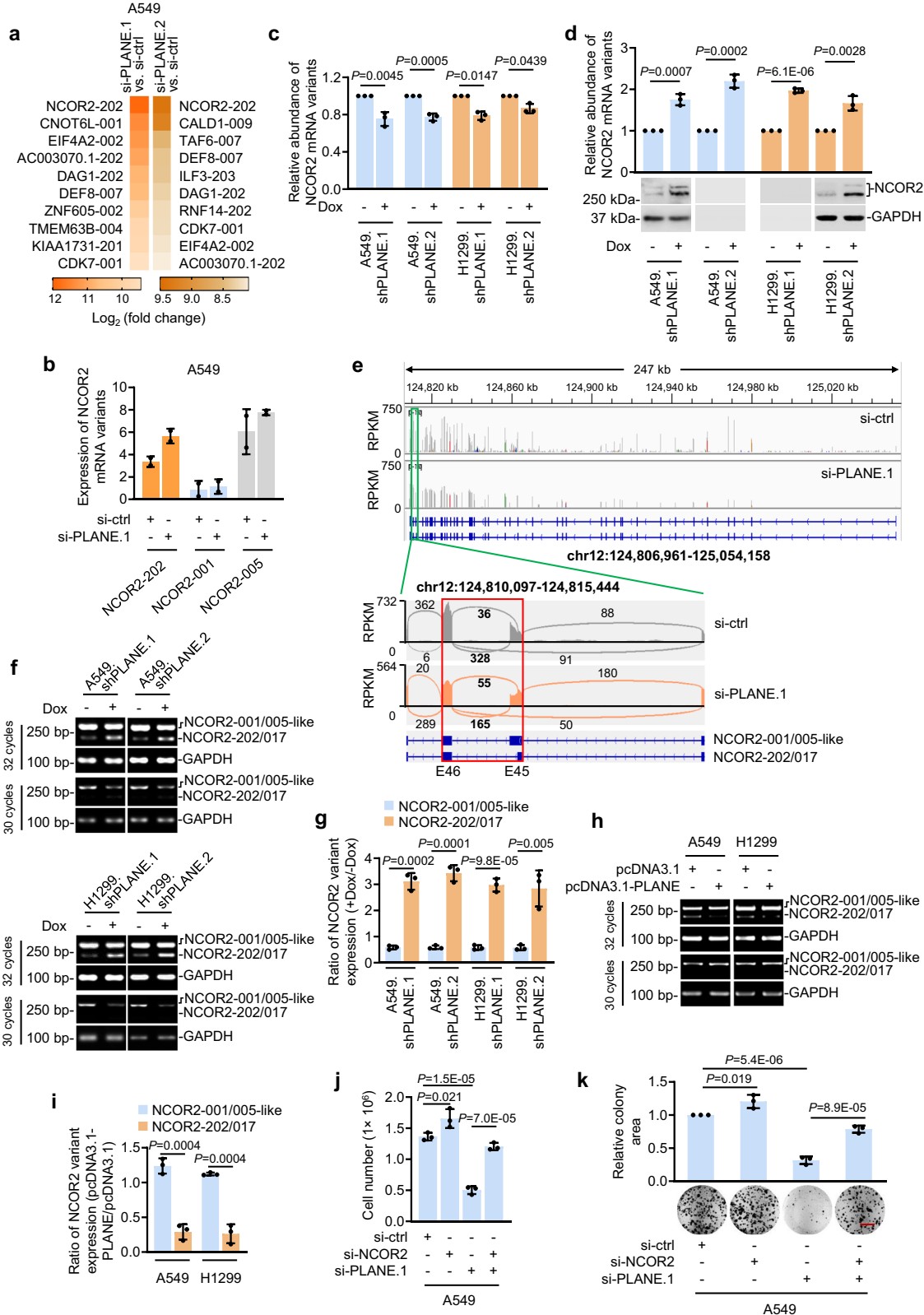

Supplementary Fig. 8d). We hereafter refer to these siRNAs as NCOR2 siRNAs for simplicity. As anticipated, introduction of the NCOR2 siRNAs diminished the upregulation of NCOR2 and reversed, at least partially, the reduction in cell proliferation and clonogenicity caused by knockdown of PLANE (Fig. 3j, k and Supplementary Fig. 9d). Of note, NCOR2 knockdown alone

caused moderate increases in cell proliferation (Fig. 3j, k and Supplementary Fig. 9d).

We lastly tested whether PLANE impinges on AS of other pre-mRNAs through analysing RNA-seq data from A549 cells using the modelling alternative junction inclusion quantification (MAJIQ) that detects and quantifies local splicing variations

**Fig. 3 PLANE represses NCOR2-202-generating AS event. a** Short-read RNA-seq data showing the top 10 commonly upregulated transcripts caused by PLANE siRNA knockdown in A549 cells. $n = 2$ experimental repeats. **b** Long-read RNA-seq data showed that the expression levels of NCOR2-202 but not NCOR2-001 and -005 were significantly increased by PLANE siRNA knockdown. $n = 2$ experimental repeats. **c** qPCR analysis using primers recognising the 1-138 fragment of intron 45 that is not contained by NCOR2-202 showing a moderately decrease of NCOR2-001 and -005 as well as -002, -018, -201 and -203 after PLANE knockdown. Data are mean ± s.d.; $n = 3$ independent experiments, two-tailed Student's $t$-test. **d** qPCR analysis using primers targeting a common region of NCOR2-001, -202, -005 as well as -002, -201 and -203 and Western blotting analysis using an antibody against NCOR2-3, -1 and -2 protein isoforms encoded individually by NCOR2-001, -202 and -005 showing upregulation of NCOR2 at the mRNA and protein levels, respectively, by PLANE knockdown. Data are representatives or mean ± s.d.; $n = 3$ independent experiments, two-tailed Student's $t$-test. **e** RNA-seq read density was computed by integrated genome viewer (IGV). Quantitative visualisation using Sashimi plot revealed that the frequency of the NCOR2-202 and -017 AS event was increased, whereas the frequency of the NCOR2-001/005-like AS event was reduced after PLANE knockdown. **f–i** Induced knockdown of PLANE promoted the NCOR2-202- and -017 AS event and reduced the NCOR2-001/005-like AS event (**f, g**), whereas PLANE overexpression reduced the NCOR2-202- and -017 AS event and moderately increased the NCOR2-001/005-like AS event (**h, i**), as shown in RT-PCR analysis and the relative quantification. Data are representatives or mean ± s.d.; $n = 3$ independent experiments, two-tailed Student's $t$-test. **j, k** Co-knockdown of NCOR2 partially reversed knockdown of PLANE-induced inhibition of A549 cell proliferation as shown in cell counting (**j**) and clonogenicity (**k**). Data are representatives or mean ± s.d.; $n = 3$ independent experiments, one-way ANOVA followed by Tukey's multiple comparison test. Scale bar, 1 cm. Source data are provided as a Source data file.

(LSVs), including classical binary splicing events and non-classical binary splits and splits involving more than two junctions[40]. MAJIQ analysis identified 46916 and 48635 AS events across 10187 genes in cells with and without PLANE knockdown, respectively, with 55 significant LSVs apart from the NCOR2-202-generating splicing event triggered by knockdown of PLANE (delta PSI ≥ 20%; confidence threshold >95%) (Supplementary Fig. 10a). By use of semi-quantitative RT-PCR we validated 5 randomly selected LSVs caused by knockdown of PLANE (Supplementary Fig. 10b). Together, these results demonstrate that PLANE regulates AS of many other pre-mRNAs in addition to NCOR2, although its role in promoting cell proliferation is largely attributable to its modulatory effects on NCOR2 pre-mRNA AS.

**PLANE forms an RNA–RNA duplex with the NCOR2 pre-mRNA.** PLANE predominantly localized to the nucleus as shown by ISH analysis of A549 cells grown on coverslips and qPCR analysis of subcellular fractions (Fig. 4a, b), suggesting the possibility through forming RNA–RNA duplexes with pre-mRNAs to regulate AS. Bioinformatics analysis using the IntaRNA program (http://rna.informatik.uni-freiburg.de) identified a potential PLANE-binding region (PLANE-BR) at intron 45 of the NCOR2 pre-mRNA that complements to a fragment enriched of duplex-forming oligonucleotides (DFO) contained in PLANE (Supplementary Fig. 11a)[41]. To test whether PLANE forms RNA–RNA duplexes with the NCOR2 pre-mRNA, we employed a cell-free assay system. In vitro-synthesized biotin-labelled PLANE precipitated an RNA fragment containing the PLANE-BR at intron 45 of the NCOR2 pre-mRNA (Fig. 4c). However, this association was diminished when the PLANE-BR or the DFO within PLANE were deleted (Fig. 4c). Moreover, biotin-labelled PLANE failed to precipitate a fragment of the NCOR2 pre-mRNA that did not contain the PLANE-BR (Fig. 4d). Consistently, in vitro-synthesized biotin-labelled PLANE also precipitated the NCOR2 pre-mRNA from A549 and H1299 cell nuclear extracts (Fig. 4e). In addition, endogenous PLANE bound to the PLANE-BR but not a non-PLANE-BR-containing fragment of endogenous NCOR2 pre-mRNA as shown in domain-specific chromatin isolation by RNA purification (dChIRP) assays (Fig. 4f). Collectively, these results reveal the formation of an RNA–RNA duplex between PLANE and the NCOR2 prem-mRNA through the DFO and PLANE-BR, respectively. Of note, treatment of nuclear extracts from A549 cells with proteinase K did not disrupt the RNA–RNA duplex formed by PLANE and the NCOR2 pre-mRNA (Fig. 4g), demonstrating the binding between PLANE and the NCOR2 pre-mRNA is direct and not protein-dependent.

We then examined the functional significance of the RNA–RNA duplex in PLANE-mediated regulation of NCOR2 pre-mRNA AS and cell proliferation. In contrast to over-expression of wild-type PLANE (Fig. 3h, i and Supplementary Fig. 5b), introduction of a PLANE mutant lacking the DFO into A549 and H1299 cells had no effect on the NCOR2-202-generating splicing event and cell proliferation (Fig. 4h, i and Supplementary Fig. 11b). Moreover, introduction of a shRNA-resistant PLANE mutant (PLANE-R) inhibited the AS event caused by PLANE knockdown (Fig. 4j, k). Taken together with preceding data (Fig. 4c–k and Supplementary Fig. 11b), these findings indicate that the formation of the RNA–RNA duplex is required for the PLANE effects on NCOR2 pre-mRNA AS and cell proliferation.

**PLANE interacts with hnRNPM.** We also interrogated the proteins that interact with PLANE using RNA-pulldown followed by mass spectrometry. The most abundant protein that co-precipitated with PLANE was hnRNPM (Fig. 5a and Supplementary Table 5), one of the hnRNP proteins that complex with heterogeneous nuclear RNA and are essential in regulating mRNA maturation processes including pre-mRNA splicing[42,43]. The association between PLANE and hnRNPM was readily confirmed using RNA pulldown and RNA immunoprecipitation (RIP) assays (Fig. 5b, c). In contrast, no association was detected between PLANE and hnRNPK that was included as a control (Fig. 5b). Similarly, there was no association between hnRNPM and the mitochondrial lncRNA lncCyt b included as an additional control (Fig. 5c). In support of the direct interaction between PLANE and hnRNPM, in vitro-synthesized PLANE co-precipitated recombinant hnRNPM in a cell-free system (Fig. 5d).

To define the region of PLANE responsible for its interaction with hnRNPM, we carried out mapping experiments with PLANE mutants transcribed in vitro (Supplementary Fig. 12a). This analysis showed that PLANE fragment 331–751 but not fragment 1–330 or 752–951 was co-precipitated with hnRNPM (Fig. 5e), indicating that the 331–751 region of PLANE is required for its association with hnRNPM. We also conducted mapping experiments to identify the structural determinant for the binding of hnRNPM with PLANE. hnRNPM contains three RNA recognition motifs (RRMs) that are located at aa 72–147, aa 206–279 and aa 654–730, respectively (Supplementary Fig. 12b). Deletion of the aa 206–279 RRM but not the aa 72–147 or aa 206–279 RRM diminished the association between hnRNPM and PLANE (Fig. 5f), indicating that the aa 206–279 RRM of hnRNPM is necessary for its binding to PLANE.

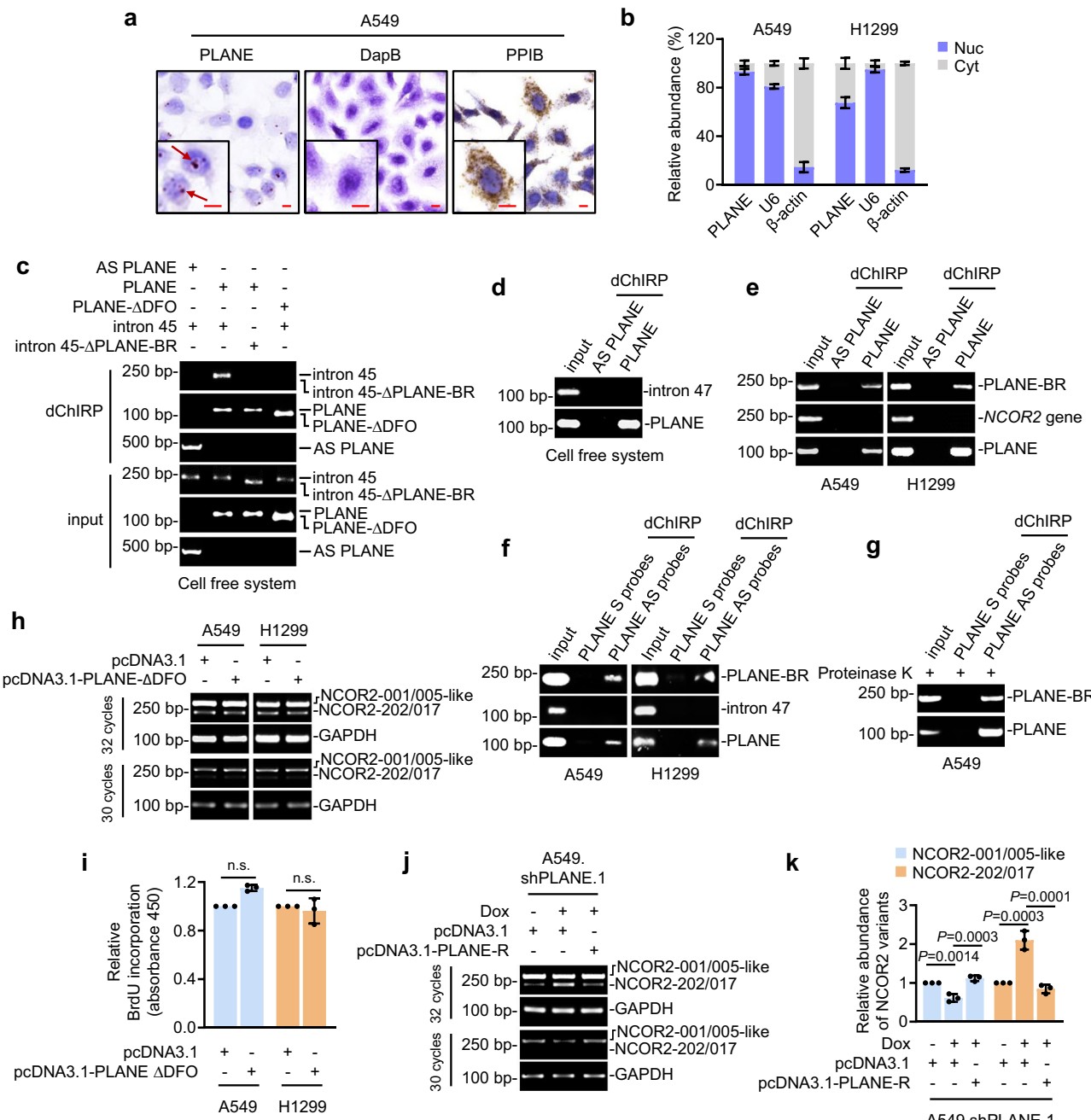

**Fig. 4 PLANE forms an RNA–RNA duplex with the NCOR2 pre-mRNA. a** Representative microphotographs of in situ hybridization (ISH) analysis of PLANE expression. DapB: negative control; PPIB: positive control. Scale bar, 10 μm. Data are representatives of 3 independent experiments. **b** qPCR analysis showing PLANE subcellular localization. Cyt: cytoplasm; Nuc: nucleus. β-actin: Cyt marker; U6: Nuc marker. Data are mean ± s.d.; *n* = 3 independent experiments, two-tailed Student's *t*-test. **c** In vitro-synthesized PLANE bound to the PLANE-binding region (PLANE-BR) of in vitro-transcribed intron 45 of NCOR2 pre-mRNA as shown in domain-specific chromatin isolation by RNA purification (dChIRP) assays. This binding was abolished when the PLANE-BR at NCOR2 pre-mRNA or the duplex-forming oligonucleotides (DFO) within PLANE were deleted (intron 45-ΔPLANE-BR and PLANE-ΔDFO, respectively). Data are representatives of three independent experiments. **d** In vitro-transcribed PLANE did not precipitate in vitro-transcribed intron 47 of the NCOR2 pre-mRNA that does not contain the PLANE-BR as shown in dChIRP assays. Data are representatives of 3 independent experiments. **e** In vitro-synthesized PLANE precipitated the endogenous NCOR2 pre-mRNA but not *NCOR2* genomic DNA as shown in dChIRP assays. Data are representatives of three independent experiments. **f** Endogenous PLANE co-precipitated a PLANE-BR-containing fragment but not a non-PLANE-BR-containing fragment of intron 47 of NCOR2 pre-mRNA as shown in dChIRP assays. Data are representatives of three independent experiments. S, sense; AS, antisense. **g** PLANE co-precipitated a fragment of intron 45 of the NCOR2 pre-mRNA in A549 cells treated with proteinase K as shown in dChIRP assays. Data are representatives of three independent experiments. **h, i** PLANE-ΔDFO did not affect the NCOR2-202-generating AS event (**h**) and 5-bromo-2'-deoxyuridine (BrdU) incorporation (**i**). Data are representatives or mean ± s.d.; *n* = 3 independent experiments, two-tailed Student's *t*-test. **j, k** A shRNA-resistant PLANE (PLANE-R) diminished the enhancement of the NCOR2-202-generating AS event caused by induced PLANE knockdown as shown in RT-PCR (**j**) and the relative quantification using densitometry (**k**). Data are representatives or mean ± s.d.; *n* = three independent experiments, one-way ANOVA followed by Tukey's multiple comparison test. Source data are provided as a Source data file.

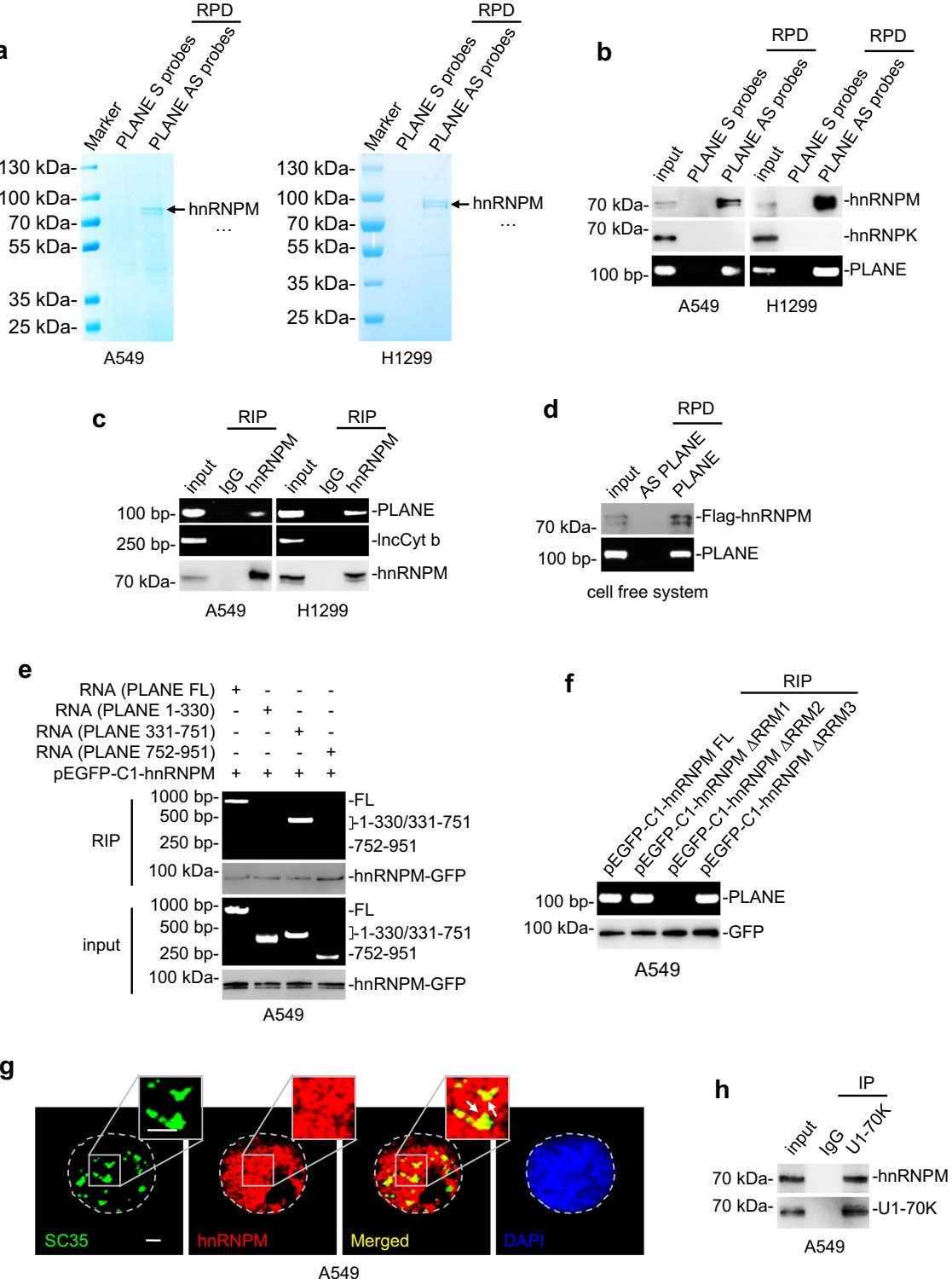

**PLANE links hnRNPM to regulation of NCOR2 pre-mRNA AS**. We next investigated the relationship of PLANE and hnRNPM in regulating NCOR2 pre-mRNA AS. As anticipated, hnRNPM predominantly localised to the nucleus in A549 and H1299 cells (Supplementary Fig. 12c). Noticeably, while a proportion of both hnRNPM and PLANE colocalised with the splicing factor SC35, a marker of nuclear speckles where the pre-mRNA splicing machinery is assembled, modified and stored[44], little NCOR2 pre-mRNA was found to colocalise with SC35 (Fig. 5g and Supplementary Fig. 12d, e). HnRNPM was also co-

precipitated with U1 small nuclear ribonucleoprotein 70 kDa (snRNP70; as known as U1-70K) that associates with the spliceosome small nuclear RNA (snRNA) U1 and is commonly used as a marker of spliceosomes (Fig. 5h)[25,45], consistent with its role as a splicing factor.

Through in silico analysis we identified multiple fragments enriched of consensus hnRNPM-binding sites (hnRNPM-BSs) in the NCOR2 pre-mRNA sequence including one site within intron 45 (Supplementary Fig. 12f and Supplementary Data 3). The latter together with three randomly selected fragments

**Fig. 5 PLANE interacts with hnRNPM. a** RNA pulldown followed by mass spectrometry analysis identified that hnRNPM is the most abundant protein co-pulled down with PLANE antisense probes in A549 and H1299 cells. S: sense; AS: antisense. $n = 1$ experiment. **b** hnRNPM was co-pulled down with PLNAE in A549 and H1299 cells as shown in RNA pulldown assays. hnRNPK was included as a negative control. S, sense; AS, antisense. Data are representatives of three independent experiments. **c** PLANE was co-precipitated with hnRNPM in A549 and H1299 cells as shown in RNA immunoprecipitation (RIP) assays. The lncRNA lncCyt b was included as a negative control. Data are representatives of three independent experiments. **d** Recombinant Flag-tagged hnRNPM was co-pulled down with in vitro-synthesized biotin-labelled PLANE as shown in RNA pulldown assays. Data are representatives of three independent experiments. **e** In vitro-synthesized full-length (FL) PLANE and PLANE fragment 331–751 but not 1–330 or 752–951 were co-precipitated with hnRNPM as shown in RIP assays. Data are representatives of three independent experiments. **f** PLANE was co-precipitated with full-length (FL) hnRNPM, hnRNPM Δ RNA recognition motif (RRM) 1 and hnRNPM ΔRRM3 but not hnRNPM ΔRRM2 as shown in RIP assays. Data are representatives of three independent experiments. **g** Representative microscopic photographs of immunofluorescence staining showing co-localization of hnRNPM and SC35 in A549 cells grown on coverslips. Data shown are representatives of three independent experiments. Scale bar: 2 μm. **h** hnRNPM was co-precipitated with U1-70K in A549 cells. Data are representatives of three independent experiments. IP: immunoprecipitation. Source data are provided as a Source data file.

containing hnRNPM-BSs (105067–105217, 158355–158512 and 176814–177141) were evaluated in RNA pulldown and/or RIP assays. Notably, all fragments except 105067–105217 were co-precipitated with hnRNPM (Fig. 6a, b and Supplementary Fig. 12g). Instructively, knockdown of PLANE reduced the relative amounts of hnRNPM binding to the hnRNPM-BSs within intron 45 but did not affect hnRNPM associations with the 158355–158512 and 176814–177141 fragments (Fig. 6c and Supplementary Fig. 12h). These observations suggest that PLANE facilitates specifically the binding of hnRNPM to the hnRNPM-BSs within intron 45 of the NCOR2 pre-mRNA. In contrast, hnRNPM did not appear to be necessary for the interaction between PLANE and the hnRNPM-BSs within intron 45, as knockdown of hnRNPM did not affect the association between PLANE and the fragment (Supplementary Fig. 12i).

Instructively, knockdown of hnRNPM enhanced the NCOR2-202 generating AS event (Fig. 6d, e), recapitulating the effect of knockdown of PLANE (Fig. 3f, g), whereas hnRNPM over-expression reduced the AS event, which was however diminished by PLANE knockdown (Fig. 6f, g), suggesting that hnRNPM is involved in PLANE-mediated regulation of NCOR2 pre-mRNA AS. Consolidating the role of PLANE in the interaction between hnRNPM and the hnRNPM-BSs, restoration of the NCOR2-202 generating AS event by introduction of PLANE-R into cells with endogenous PLANE knockdown was associated with reinstatement of the association between hnRNPM and the hnRNPM-BSs (Figs. 4j, k and 6c). However, introduction of a PLANE mutant with the 331–751 fragment deleted to disrupt its interaction with hnRNPM or with its DFO deleted to interfere with its interaction with the NCOR2 pre-mRNA did not restore the hnRNPM-hnRNPM-BS association (Fig. 6h). Collectively, these results indicate that PLANE facilitates the binding of hnRNPM with the NCOR2 pre-mRNA and is necessary for hnRNPM-mediated regulation of NCOR2 pre-mRNA AS.

Finally, as pre-mRNA splice and gene transcription frequently occur concurrently at the same sub-nuclear location[46,47], it is possible that PLANE similarly navigates hnRNPM locally to the NCOR2 gene. However, in contrast to the NCOR2 pre-mRNA, the NCOR2 gene was not co-precipitated by in vitro-synthesized biotin-labelled PLANE in nuclear extracts from A549 and H1299 cell nuclear extracts (Fig. 4e), indicating that PLANE does not have a direct role in promoting the association, if any, between hnRNPM and the NCOR2 gene.

## Discussion

A number of proteins encoded by genes located to the distal portion of chromosome 3q that is frequently amplified in various cancer types are known to drive cancer pathogenesis, such as the p110α subunit of phosphatidylinositol 3-kinase (PI3Kα) and

eukaryotic translation initiation factor 4G (eIF4G)[48,49]. In this study, we demonstrate that PLANE, a lncRNA encoded by a gene situated in this region, is similarly upregulated in diverse cancer types and promotes cancer cell proliferation and tumorigenicity, thus uncovering a hitherto unrecognised oncogenic contribution of a non-protein-coding component of the distal portion of chromosome 3q. Nevertheless, genomic amplification is not the only mechanism responsible for the increased PLANE expression in cancer cells, rather, PLANE upregulation is more commonly driven by E2F1-mediated transcriptional activation. As a transcription factor with dichotomous functions, E2F1 on one hand transactivates many protein-coding genes involved in cell cycle progression and its high expression causes tumorigenesis[50,51], but on the other hand, E2F1 loss has also been demonstrated to induce cancer development and progression[52]. Our results identified transcriptional activation of PLANE as a mechanism involved in E2F1 promotion of cell proliferation, suggesting that PLANE may represent a potential target for counteracting the cancer-promoting axis of E2F1 signalling.

PLANE promoted cancer cell proliferation and tumorigenicity through inhibition of the expression of NCOR2, which, as a transcriptional co-repressor, functions by way of a platform that links chromatin-modifying enzymes such as HDACs and transcription factors to regulate transactivation of downstream genes involved in many cellular processes including cell survival and proliferation[9–11]. As such, deregulation of NCOR2 is associated with the pathogenesis of various diseases including cancer[14–18]. In support of our results, a number of studies have demonstrated a tumour suppressive role of NCOR2 in cancers, such as LUAD, head and neck squamous cell carcinoma, non-Hodgkin lymphoma and osteosarcoma[14–17]. However, NCOR2 has also been shown to promote cell survival and proliferation in some other cancer types such as breast cancer[18], suggestive of an oncogenic role. These paradoxical observations are nevertheless consistent with the notion that NCOR2 functions in a manner closely related to the diverse repertoire of NCOR2 isoforms generated by AS at its C-terminal interaction domains resulting in varying affinities for different transcription factors[9,12,13]. Indeed, we found that PLANE-mediated suppression of NCOR2 expression was due to selective repression of AS production of the NCOR2-202 transcript that encodes one of the major NCOR2 protein isoforms, NCOR2 isoform 2[36], demonstrating a tumour suppressive function of this isoform. Nonetheless, whether each of the other NCOR2 isoforms has any specific effect on cancer pathogenesis remains to be defined. The expression of the NCOR2 isoform BQ323636.1 is known to confer chemoresistance in breast cancer[53]. Moreover, the NCOR2 pre-mRNA splicing pattern may change in a context-dependent fashion as it does during adipocyte differentiation[54]. Of note, PLANE is the major isoform of MELTF-AS1/MFI2-AS1 that has been reported to play

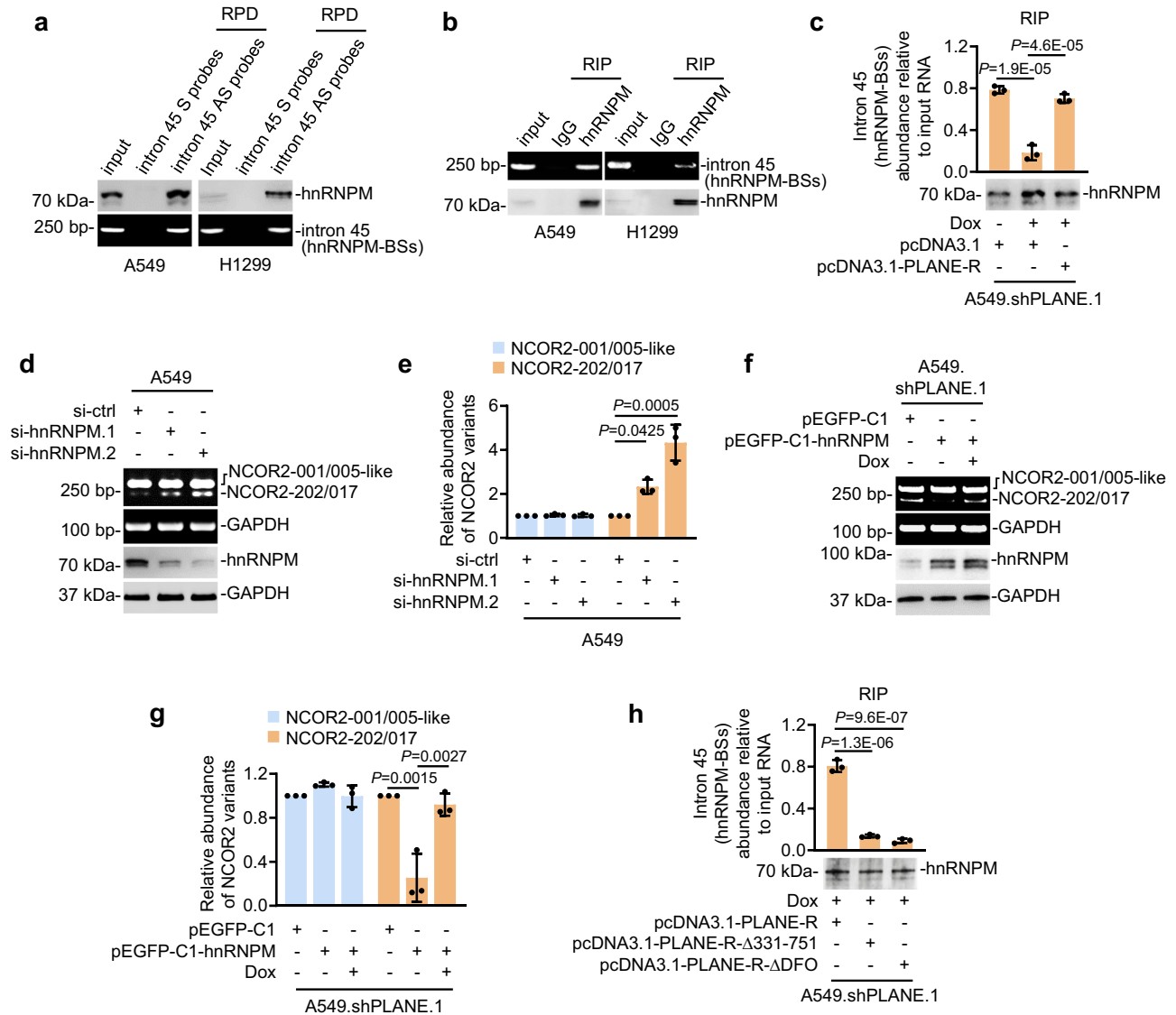

**Fig. 6 PLANE links hnRNPM to regulation of NCOR2 pre-mRNA AS. a** hnRNPM was co-pulled down with the NCOR2 pre-mRNA using antisense probes directed to the hnRNPM-binding sites (hnRNPM-BSs) at intron 45 in A549 and H1299 cells as shown in RNA pulldown assays. Data are representatives of three independent experiments. S, sense; AS, antisense. **b** The hnRNPM-BSs at intron 45 of the NCOR2 pre-mRNA was co-precipitated with hnRNPM in A549 and H1299 cells using RIP assays. Data are representatives of three independent experiments. **c** Induced knockdown of PLANE decreased the amount of hnRNPM associated with the hnRNPM-BSs at the NCOR2 pre-mRNA, which was reversed by co-overexpression of a shRNA-resistant PLANE mutant (PLANE-R) as shown in RIP assays. Data are representatives or mean ± s.d.; $n = 3$ independent experiments, one-way ANOVA followed by Tukey's multiple comparison test. **d** SiRNA knockdown of hnRNPM enhanced the NCOR2-202-generating AS event in A549 cells. Data are representatives of three independent experiments. **e** Relative levels of NCOR2-202/017 and NCOR2-001/005-like mRNA variants in cells with or without hnRNPM knockdown as shown in (**d**). Data are mean ± s.d.; $n = 3$ independent experiments, one-way ANOVA followed by Tukey's multiple comparison test. **f** Overexpression of hnRNPM reduced the NCOR2-202-generating AS event, which was reversed by co-knockdown of PLANE in A549 cells. Data shown are representatives of 3 independent experiments. **g** Relative levels of NCOR2-202/-001/-005 as shown in (**f**) quantitated using densitometry. Data are mean ± s.d.; $n = 3$ independent experiments, one-way ANOVA followed by Tukey's multiple comparison test. **h** Introduction of a shRNA-resistant PLANE mutant (PLANE-R) but not a PLANE mutant with its fragment 331–751 deleted (PLANE-R-Δ331-751) or its DFO deleted (PLANE-R-ΔDFO) induced the association between hnRNPM and the hnRNPM-BSs at the NCOR2 pre-mRNA in A549.shPLANE cells with induced knockdown of PLANE. Data are representatives or mean ± s. d.; $n = 3$ independent experiments, one-way ANOVA followed by Tukey's multiple comparison test. Source data are provided as a Source data file.

an oncogenic role in multiple cancer types[55,56]. Although these studies did not specify which isoform(s) were investigated, we anticipate they would likely have examined the most abundant isoform that we now call PLANE.

Mechanistic investigations revealed that the formation of an RNA–RNA duplex was necessary for PLANE to repress the AS event generating NCOR2-202, acting through the DFO within PLANE and the complementary PLANE-BR at intron 45 of

NCOR2 pre-mRNA. The nature of the duplex interaction was substantiated through several independent experimental approaches and importantly it was demonstrated that disruption of the duplex was sufficient to abolish the repression of the NCOR2-202-generating AS event and inhibition of cell proliferation. A number of lncRNAs have been shown to regulate AS through forming RNA–RNA duplexes with pre-mRNAs and thus regulate gene expression through modification of the transcript

landscape[24,28,57]. This is commonly accomplished through targeting particular splicing factors by lncRNA to selective cis-acting nucleotide sequences[24]. For example, the lncRNA SAF complexes with the Fas pre-mRNA at the exon 5–6 and exon 6–7 junction and binds to SPF45, thus facilitating exclusion of exon 6 leading to the generation of a Fas protein isoform that lacks the trans-membrane domain and cannot induce apoptosis[28]. Moreover, the lncRNA zinc finger E-box binding homeobox 2 (ZEB2) antisense RNA 1 (ZEB2-AS1) forms RNA–RNA duplex with the ZEB2 pre-mRNA and thus prevents the recognition of the spliceosome resulting in intron inclusion at the ZEB 5′-UTR and consequent promotion of ZEB2 translation, activating EMT[57]. Likewise, the RNA–RNA duplex formed by PLANE and the NCOR2 pre-mRNA functions to facilitate the association of hnRNPM with intron 45 of the pre-mRNA leading to suppression of the NCOR2-202 generating AS event. Of note, PLANE was also found to regulate many other AS events, although the biological consequences remain to be defined. Similarly, the mechanisms regulating the formation of RNA–RNA duplex between PLANE and selective pre-mRNAs remain to be determined. Regardless, our results demonstrate that the effect of PLANE on cancer cell proliferation is associated with regulation of the NCOR2-202-generating AS event. Nonetheless, while our data rule out the possibility that PLANE serves as an agent supporting the nuclear export of NCOR2 mRNA variants, other prospective mechanisms remain open. Given its predominant nuclear localization, further studies to clarify whether PLANE regulates NCOR2 mRNA maturation and stability are clearly warranted.

Like other hnRNP family proteins, hnRNPM is involved in regulating RNA maturation processes[42,43]. In particular, it controls AS of a variety of pre-mRNAs of cancer-associated genes[42,58–60]. For instance, hnRNPM mediates AS of fibroblast growth factor receptor 2 (FGFR2) and CD44 pre-mRNAs to promote EMT[58–60]. Moreover, hnRNPM guides an AS program involving multiple pre-mRNAs to confer resistance of Ewing sarcoma cells to inhibition of the PI3K/AKT/mTOR signal pathway[25]. Consistent with these cancer-promoting actions, our results showed that hnRNPM along with PLANE represses the AS event generating NCOR2-202 and thus promotes cancer cell proliferation. Although hnRNPs commonly act as splice suppressors when binding to exonic motifs, and as splicing enhancers when associating with motifs in introns[61,62], its binding to the hnRNPM-BSs at intron 45 of the NCOR2 pre-mRNA results in suppression of the NCOR2-202-generating AS event, suggesting that hnRNPM can function as an AS suppressor when it binds to intronic regulatory sequences. The finding that PLANE links hnRNPM to repress the NCOR2-202-generating AS event indicates that PLANE determines the selectivity of hnRNPM on the splice site of the NCOR2 pre-mRNA. Indeed, there is increasing evidence showing that lncRNAs can function as "local address codes" to target regulatory proteins to nucleotide sequences[63].

The hnRNPM protein is highly abundant in cells[25,64]. A previous quantitative proteomic study in U2OS cells showed that hnRNPM was present with >1,400,000 molecules[64]. On the other hand, PLANE is expressed at markedly lower abundance, ~157–262 molecules per A549 and H1299 cells. A question arising from this disparity is whether the limited numbers of PLANE molecules are sufficient to link enough hnRNPM necessary for repressing the NCOR2-202-generating AS event. Nevertheless, hnRNP family members make multi-faceted contributions to nucleic acid metabolism and hnRNPM is not only involved in the regulation of NCOR2 pre-mRNA alternative splicing[42]. Thus, the abundance of hnRNPM reflects its vastly broader regulatory roles in cells compared to PLANE. Moreover, the function of hnRNPM as a splice factor is tightly regulated by its posttranslational modifications, for example, phosphorylation

and sub-nuclear translocation[43]. Indeed, the vast majority of hnRNPM was found to locate to the nuclear speckles, where the pre-mRNA splicing machinery is assembled, modified and stored[44], whereas the execution of a specific AS event is highly dynamic and conceivably requires limited resources from the general splicing machinery[44]. It is therefore conceivable that the factual difference between the number of PLANE copies and the number of hnRNPM molecules is conceivably not as large as estimated at the face value. Regardless, that PLANE links hnRNPM to repression of the NCOR2-202-generating AS is not in dispute since the process can be modelled in vitro and the action is sufficient to suppress cancer cell survival tumorigenicity. Our current evidence suggests PLANE does not have a direct role in promoting association between hnRNPM and DNA since the NCOR2 gene was not captured with PLANE in nuclear extracts.

One of the features of lncRNAs compared to protein-coding genes is their relatively poor sequence conservation[29,63]. By use of bioinformatics analysis, we identified a Gorilla transcript that is highly homologous to human PLANE with 92% sequence similarity (Supplementary Table 6), suggesting evolutionary conservation of PLANE between Hominidae. Nonetheless, no similarity was found between PLANE and Mus musculus, a finding that precluded further testing the role of PLANE in transgenic mouse models (Supplementary Table 6). Irrespectively, our results from functional and correlative studies using human cell line models and human tissue samples suggest that PLANE contributes to cancer development and progression driven by genomic amplification of the distal portion of chromosome 3q and the cancer-promoting axis of E2F1 signalling (Supplementary Fig. 13). PLANE may therefore represent a potential anti-cancer target for countering these oncogenic anomalies.

## Methods

**Cell culture and human tissues.** A549, MCF-7, HCT116, Eca109 and CCC-HIE-2 cells were maintained in DMEM (Biological Industries, #01-052-1ACS; Beit Haemek, Israel) supplemented with 10% foetal bovine serum (FBS, Biological Industries, #04-001-1A; Beit Haemek, Israel) and 1% penicillin-streptomycin (Biological Industries, #03-031-1BCS, Beit Haemek, Israel). H1299, NCI-H1975 and NCI-H226 cells were cultured in RPMI-1640 (Biological Industries, #01-100-1ACS; Beit Haemek, Israel) with 10% FBS and 1% penicillin-streptomycin. CCC-HSF-1 cells were cultured in DMEM/F12 (Biological Industries, #01-172-1ACS; Beit Haemek, Israel) supplemented with 10% FBS and 1% penicillin-streptomycin. Cells were cultured in a humidified incubator at 37 °C and 5% $CO_2$. All cell lines were verified to be free of mycoplasma contamination every 3 months. Individual cell line authentication was confirmed using the AmpFISTR Identifiler PCR Amplification Kit (ThermoFisher Scientific, #4427368) from Applied Biosystems and Gene-Marker V1.91 software (SoftGenetics LLC). Information on cell lines is provided in Supplementary Table 7. Studies using formalin-fixed paraffin-embedded (FFPE) normal colon mucosa, colon adenoma, and COAD tissues retrieved from archives of the Department of Pathology at Shanxi Cancer Hospital (Taiyuan, China) were approved by the Human Research Ethics Committees of the Shanxi Cancer Hospital. Studies using freshly isolated LUAD and LUSC and adjacent normal tissues collected by the Henan Provincial People's Hospital (Zhengzhou, China) were approved by the Human Research Ethics Committees of the Henan Provincial People's Hospital. Studies using freshly isolated BRCA and adjacent normal tissues collected by the Department of Breast Surgery, Shanxi Bethune Hospital (Taiyuan, China) were approved by the Human Research Ethics Committees of Shanxi Bethune Hospital. Studies using freshly isolated ESCC and adjacent normal tissues collected by the Department of Thoracic Surgery, the First Affiliated Hospital of Anhui Medical University (Hefei, China) were approved by the institutional review board of Anhui Medical University.

**Antibodies and reagents.** Information on antibodies and reagents used in this study is provided in Supplementary Tables 8 and 9, respectively.

**SiRNAs and short hairpin RNA (shRNA) Oligos.** SiRNAs were obtained from GenePharma (Shanghai, China) and transfected using the lipofectamine 3000 Transfection Kit (ThermoFisher Scientific, #L3000-015). ShRNA oligos were purchased from TSINGKE Biological Technology (Beijing, China).

**Plasmids.** The FH1-tUTG plasmid was a kind gift from A/Professor M. J. Herold (Walter and Eliza Hall Institute of Medical Research, Australia). The pcDNA3.1

(+), pGL4.73[hRluc/SV40] and pSin-3 × Flag-E2F1 plasmids were kind gifts from Professor Mian Wu (Translational Research Institute, Henan Provincial People's Hospital and People's Hospital of Zhengzhou University, Zhengzhou, China). The pEGFP-C1 plasmid was a kind gift from A/Professor Yongyan Wu (Department of Otolaryngology, Shanxi Key Laboratory of Otorhinolaryngology Head and Neck Cancer, the first affiliated hospital, Shanxi Medical University, Taiyuan, China). The pMDLg/pRRE plasmid (#12251), pMD2.g plasmid (#12259) and pRSV-Rev plasmid (#12253) were purchased from Addgene. The pGL3-*PLANE*-promoter and the pGL3-*PLANE*-promoter-ΔE2F1-BR were purchased from Sangon Biotech (Shanghai, China). Other plasmids used in this study were generated by inserting the PCR products to the pcDNA3.1(+) or pEGFP-C1 vectors. Primers used in the fusion PCR are shown in Supplementary Table 10.

**Short-read RNA sequencing (RNA-seq)**. Short-read RNA-seq was conducted by GENEWIZ (Suzhou, China). 1 μg RNA per sample was used for library preparation according to the manufacturer's protocol. The Poly(A) mRNA isolation was performed using Poly(A) mRNA Magnetic Isolation Module or rRNA removal Kit. The mRNA fragmentation and priming were performed using First Strand Synthesis Reaction Buffer and random primers. First-strand cDNA was synthesized using ProtoScript II Reverse Transcriptase and the second-strand cDNA was synthesized using Second Strand Synthesis Enzyme Mix. The purified double-stranded cDNA by beads was then treated with End Prep Enzyme Mix to repair both ends and add a dA-tailing in one reaction, followed by a T-A ligation to add adaptors to both ends. Size selection of adaptor-ligated DNA was then performed using beads, and fragments of ~400 bp (with the approximate insert size of 300 bp) were recovered. Each sample was then amplified by PCR using P5 and P7 primers, with both primers carrying sequences which can anneal with flow cell to perform bridge PCR and P5/ P7 primer carrying index allowing for multiplexing. After bead cleanup, the PCR products were validated using an Qsep100 (Bioptic, Taiwan, China), and quantified by Qubit3.0 Fluorometer (Invitrogen, Carlsbad, CA, USA).

Then the libraries were sequenced using an Illumina HiSeq 4000 platform according to manufacturer's instructions (Illumina, San Diego, CA, USA) to generate 2 × 150 paired-end (PE) reads. Image analysis and base calling were conducted by the HiSeq Control Software (HCS) + OLB + GAPipeline-1.6 (Illumina) on the HiSeq instrument, image analysis and base calling were conducted by the NovaSeq Control Software (NCS) + OLB + GAPipeline-1.6 (Illumina) on the NovaSeq instrument, image analysis and base calling were conducted by the Zebeacall on the MGI2000 instrument. Differential expression analysis used the DESeq2 Bioconductor package, a model based on the negative binomial distribution. the estimates of dispersion and logarithmic fold changes incorporate data-driven prior distributions, Padj of genes were set <0.05 to detect differential expressed ones.

To determine alternative splicing events, Asprofile v1.0 was employed to analyze all pairs of transcripts to determine exons included in one transcript and spliced in another from RNA-seq data[65]. Briefly, short reads were mapped to the human genome and then assembled overlapping reads on the genome into transcript fragments using Cufflinks[66]. Cufflinks represents all reads at a locus as an assembly graph, in which any two reads are connected if they overlap and have compatible splice patterns, and then traverses the graph to produce the minimum number of transcripts that can explain all of the input reads. Because single-exon transcripts, which form the bulk of the assemblies, are frequently artifacts of sequencing and mapping, we used only the multiexon transcripts to measure the gene and transcript content. The short-reads RNA-seq data were deposited in the Gene Expression Omnibus (GEO) under accession code GSE162215.

**Long-read RNA-seq**. Long-read RNA-seq was conducted by Biomarker Technologies (Beijing, China). Briefly, 1 μg total RNA was prepared for cDNA libraries using cDNA-PCR Sequencing Kit (SQK-PCS109) protocol provided by Oxford Nanopore Technologies (ONT). The template switching activity of reverse transcriptase enrich for full-length cDNAs and add defined PCR adaptors directly to both ends of the first-strand cDNA. And following cDNA PCR for 14 circles with LongAmp Taq (NEB). The PCR products were then subjected to ONT adaptor ligation using T4 DNA ligase (NEB). Agencourt XP beads was used for DNA purification according to ONT protocol. The final cDNA libraries were added to FLO-MIN109 flowcells and run on the PromethION platform.

Raw reads were first filtered with minimum average read quality score = 7 and minimum read length = 500 bp. Ribosomal RNA was discarded after mapping to rRNA database. Next, full-length, non-chemeric (FLNC) transcripts were determined by searching for primer at both ends of reads. Clusters of FLNC transcripts were obtained after mapping to reference genome with mimimap2, and consensus isoforms were obtained after polishing within each cluster by pinfish. Consensus sequences were mapped to reference genome using minimap2. Mapped reads were further collapsed by cDNA_Cupcake package with min-coverage = 85% and min-identity = 90%. A 5′ difference was not considered when collapsing redundant transcripts. The criteria for fusion candidates are that a single transcript must (1) map to 2 or more loci; (2) minimum coverage for each locus is 5% and minimum coverage in bp is ≥1 bp; (3) total coverage is ≥95%; (4) distance between the loci is at least 10 kb.

Full-length reads were mapped to the reference transcriptome sequence. Reads with match quality above 5 were further used to quantify. Expression levels were estimated by reads per gene/transcript per 10,000 reads mapped. Differential expression analysis of two conditions/groups was performed using the DESeq2 R package (1.6.3). DESeq2 provide statistical routines for determining differential expression in digital gene expression data using a model based on the negative binomial distribution. The resulting $P$ values were adjusted using the Benjamini and Hochberg's approach for controlling the false discovery rate. The long-reads RNA-seq data were deposited in the Gene Expression Omnibus (GEO) under accession code GSE172124.

**Quantitative PCR (qPCR)**. Total RNA was extracted from cultured cells using the Gene JET RNA Purification Kit (ThermoFisher Scientific, #K0731) according to the manufacturer's instructions. cDNA was synthesized from 1 μg of total RNA using the PrimScriptTM RT reagent Kit with gDNA Eraser (TaKaRa, #RR047A; Dalian, China). Of the resultant cDNA, 12.5 ng was used in the 20 μl qPCR mix, containing 10 μl of TB Green Premix Ex Taq II (Tli RNaseH Plus) (TaKaRa, #RR820A; Dalian, China) and 0.4 μM of each primer. Samples were amplified for 40 cycles using a StepOnePlusTM Real-Time PCR System (ThermoFisher Scientific). $2^{-\Delta\Delta CT}$ method was used to calculate the relative gene expression levels normalized against an average of three housekeeping genes (GAPDH, β-actin and 18s rRNA) as previously described[67]. Primer sequences are listed in Supplementary Table 11.

**Chromatin immunoprecipitation (ChIP)**. ChIP assays were performed using the ChIP Assay Kit (Beyotime, #P2078; Shanghai, China) according to the manufacturer's instructions. Briefly, cells were cross-linked with a final concentration of 1% formaldehyde in growth medium for 15 min at 37 °C and quenched by the addition of glycine solution for 5 min at room temperature (RT). Then cells were harvested, lysed and sonicated. After being cleared by centrifugation at 12,000 × g for 10 min at 4 °C, the cell lysate was subjected to a 1:10 dilution and rotated with E2F1, H3K4me3 and H3K27me3 antibodies or corresponding mouse/rabbit normal immunoglobulin (IgG) antibodies at 4 °C overnight. Then, 60 μl of protein A/ G agarose beads was added to the antibody-lysate mixture and rotated at 4 °C for an additional 1 h. Beads were washed, and DNA fragments were eluted, purified and subjected to PCR analysis using the specific primers. PCR products were separated by gel electrophoresis on the 2% agarose gel. Information on antibodies and primers used in this study are shown in Supplementary Tables 8 and 12, respectively.

**Luciferase reporter assays**. A549 and H1299 cells were transfected with pGL3-*PLANE*-promoter reporters or pGL3-*PLANE*-promoter-ΔE2F1-BR reporters together with pGL4.73[hRluc/SV40] reporters expressing the renilla luciferase. After 48 h, firefly and renilla luciferase activities were examined by a Dual-Luciferase® Reporter Assay System (Promega, #E1910) with a VARIOSKAN LUX microplate reader. The renilla luciferase activity was used to normalize the firefly luciferase activity.

**Colony formation**. Cancer cells were seeded in six-well plates at 2000 cells/well. After growing for further 2 weeks, cells were fixed with methanol and staining with 0.5% crystal violet. The images were captured with a Bio-Rad GelDoc™ XR + imaging system (Bio-Rad). The percentage and intensity of area covered by crystal violet-stained cell colonies were quantified using ImageJ-plugin "ColonyArea". The images were cropped using adobe photoshop elements software.

**Cell cycle analysis**. Cell cycle analysis was performed using the Cell Cycle and Apoptosis Analysis Kit (Meilunbio, #MA0334; Dalian, China) according to the manufacturer's instructions followed by flow cytometry. Briefly, A549 and H1299 cells transfected with PLANE siRNAs for 48 h in 24-well plates were harvested and fixed in 75% ethanol at 4 °C overnight. After being centrifuged, cells were incubated in the staining solution at 37 °C in the dark for 30 min. Then cells were subjected to analysis using a flow cytometer (FACSAria, BD Biosciences) and BD FACS Diva v8.0.2 software. The gating strategy is shown in Supplementary Fig. 14.

**Anchorage-independent cell growth**. Cells carrying an inducible PLANE knockdown in response to Dox were seeded in the Ultra-Low attachment 6-well plate (Corning, #3471) at 2000 cells/well. Cells with or without treatment with doxycycline (Dox) and cessation of Dox treatment were incubated at 37 °C in a humidified incubator until colonies were formed. Colonies were counted under a light microscope[29].

**In situ hybridization (ISH)**. ISH assays were performed using the RNAscope® 2.5 HD Detection Reagent-BROWN (Advanced Cell Diagnostics, #322310) according to the manufacturer's instructions[63,68]. Briefly, FFPE LUSC and LUAD as well as COAD tissue microarrays (#HLug-Squ150Sur-02, #HLugA180Su03, #HCo-lA180Su12) purchased from the Shanghai Outdo Biotech Co., Ltd (China) were deparaffinized in xylene for 5 min at RT twice, followed by dehydrization in 100% alcohol. After being air-dried, the tissue sections were incubated with hydrogen peroxide for 10 min at RT and washed in the distilled water five times. Then the sections were heated in target retrieval reagent to 100 °C for 20 min, followed by being treated with proteinase K and incubated in hybridization buffer

containing probes (Advanced Cell Diagnostics, #570031) at 40 °C for 3 h. After being washed, the sections were incubated with 3,3′-diaminobenzidine (DAB), and counterstaining was carried out using hematoxylin.

The percentage of positive cells was ranged from 0 to 100%. The intensity of staining (intensity score) was judged on an arbitrary scale of 0–4: no staining (0), weakly positive staining (1), moderately positive staining (2), strongly positive staining (3) and very strong positive staining (4). A reactive score (RS) was derived by multiplying the percentage of positive cells with staining intensity divided by 10.

**Immunofluorescence (IF)**. Cells grown on coverslips were fixed in 4% formaldehyde for 10 min at RT. After being washed using PBS, cells were then permeabilized in blocking buffer for 60 min at RT. Antibodies diluted 1:500 in blocking buffer were incubated with cells overnight at 4 °C. Cells were washed in PBS and incubated with secondary antibodies diluted 1:200 in blocking buffer for 60 min at RT in the dark. After being washed, cells were mounted in the Pro-Long™ Glass Antifade Mountant with NucBlue reagent (ThermoFisher Scientific, P36981). Images were digitally recorded using a Leica SP8 confocal microscope. Information of antibodies used in this study was shown in Supplementary Table 8.

**Subcellular fractionation**. Cells were harvested by trypsinization and lysed in hypotonic buffer A (10 mM Hepes pH 7.9, 10 mM KCl, 0.1 mM EDTA, 0.1 mM EGTA, 1 mM DTT, 0.15% Triton X-100, cOmplete™, EDTA-free Protease Inhibitor Cocktail) on ice for 15 min. The supernatants after centrifugation at 12,000 × g for 3 min were collected as the cytoplasmic fractions and the pellets were subjected to the nuclear fractionation. The pellets were rinsed with cold PBS once and lysed in an equal volume of buffer B (20 mM Hepes pH 7.9, 400 mM NaCl, 1 mM EDTA, 1 mM EGTA, 1 mM DTT, 0.5% Triton X-100, cOmplete™, EDTA-free Protease Inhibitor Cocktail) on ice for 15 min. Cytoplasmic and nuclear fractions were centrifuged at 16,000 × g for 20 min to remove the insoluble debris. The supernatants were collected for RNA isolation and immunoblotting analysis.

**In vitro transcription**. The DNA templates used for in vitro synthesis of PLANE, antisense PLANE and NCOR2 pre-mRNA were generated by PCR amplification from cDNAs using PrimerSTAR Max DNA Polymerase (TAKARA, #R045A; Dalian, China). Forward primers containing the T7 RNA polymerase promoter sequence and reverse primers without the promoter sequence were used for synthesizing PLANE, antisense PLANE, and NCOR2 pre-mRNA. After PCR amplification, the products were purified using a MiniBEST Agarose Gel DNA Extraction Kit (Takara, #9762; Dalian, China), and subjected to in vitro transcription using a TranscriptAid T7 High Yield Transcription Kit (ThermoFisher Scientific, #K0441) according to the manufacturer's instructions. The in vitro-transcribed RNAs could be further labelled with biotin using a Pierce™ RNA 3′ End Desthiobiotinylation Kit (ThermoFisher Scientific, #20163). Primer sequences are shown in Supplementary Table 12.

**Domain-specific chromatin isolation by RNA purification (dChIRP)**. dChIRP assays were performed as previously described[69]. Briefly, A549 and H1299 cells were harvested and cross-linked in 1% glutaraldehyde for 10 min at RT with rotation. The cross-linked cells were lysed in lysis buffer (50 mM Tris-Cl [pH 7.0], 10 mM EDTA, 1% SDS, PMSF, Superase-in), followed by sonication. Four micrograms antisense/sense biotin-labelled probes against PLANE RNA or 10 μg in vitro-transcribed biotin-labelled PLANE RNA/PLANE antisense RNA was rotated with cell lysates at 37 °C for 4 h, followed by adding 100 μl C-1 magnetic beads (Invitrogen, #65002) to each sample and incubating at 37 °C for 30 min with rotation. Beads were then washed in wash buffer for five times, followed by RNA isolation. Probe sequences are shown in Supplementary Table 12.

**Biotin RNA pull-down (RPD)**. A549 and H1299 cells were harvested and washed in PBS three times. Cell pellets were then lysed in lysis buffer (50 mM Tris-HCl [pH 7.5], 150 mM NaCl, 2.5 mM MgCl₂, 1 mM EDTA, 10% Glycerol, 0.5% Nonidet P-40/Igepal CA-630, 1 mM DTT, cOmplete™ EDTA-free Protease Inhibitor Cocktail and RNase inhibitors) and sonicated. Four micrograms of antisense/sense biotin-labelled probes were incubated with lysates at 4 °C overnight before rotating with streptavidin beads (ThermoFisher Scientific, #20349) for additional 2 h. Beads were then washed in lysis buffer four times, followed by RNA isolation and immunoblotting analysis using Image Studio software. Information of antibodies and probes is shown in Supplementary Tables 8 and 12, respectively.

**Mass spectrometry (MS) analysis**. Proteins co-pulled down with RNA using antisense/sense biotin-labelled probes were separated by 10% acrylamide gels and visualized by Coomassie brilliant blue staining. The specific protein band shown in the group using antisense probes along with the corresponding region in the group using sense probes were resected and digested, followed by the liquid chromatography–mass spectrometry (LC-MS) analysis using a mass spectrometer (ThermoFisher Scientific, EASY-nLC1000 & LTQ Orbitrap Velos Pro). Proteins identified from the mass spectrometry analysis are listed in Supplementary Table 5.

**RNA immunoprecipitation (RIP)**. RIP assays were performed using a Magna RIP™ Kit (Millipore, #17-700; Darmstadt, Germany) according to the instruction provided by the manufacturer. Briefly, cell lysates prepared in hypotonic buffer supplemented with RNase inhibitor and protease inhibitor were incubated with magnetic beads pre-incubated with hnRNPM antibodies at 4 °C overnight. After being washed with RIP wash buffer, the bead-bound immunocomplexes were subjected to immunoblotting analysis and RNA isolation. Information on antibody and primers used in this study is shown in Supplementary Tables 8 and 12.

**Immunoprecipitation (IP)**. Cells were collected with trypsinization and lysed with lysis buffer (20 mM Tris-HCl pH 8.6, 100 mM NaCl, 20 mM KCl, 1.5 mM MgCl₂, 0.5% NP-40, cOmplete™ EDTA-free Protease Inhibitor Cocktail) on ice for 1 h and centrifuged at 16,000 × g for 30 min. After quantification using a BCA protein assay kit (ThermoFisher, #23225), 3 mg of total protein were rotated with antibodies at 4 °C overnight. Protein-antibody complexes were then captured with the Pierce™ Protein A/G Agarose (ThermoFisher Scientific, #20421) at 4 °C for 2 h with rotation and beads were then rinsed with wash buffer (25 mM Tris, 150 mM NaCl, pH 7.2), boiled and subjected to immunoblotting analysis. Antibodies used in this study are shown in Supplementary Table 8.

**Absolute quantification of PLANE**. Absolute RNA quantification was performed using the standard curve method by qPCR. cDNA was synthesized using 1 μg of the total RNA extracted from a fixed cell number. Ten-fold serial dilutions of the pcDNA3.1-PLANE plasmid (10²–10⁷ molecules per ml) were used as a reference molecule for the standard curve calculation. Assays were reconstituted to a final volume of 20 μl using 5 μl cDNA from cells or 5 μl serial diluted pcDNA3.1-PLANE plasmid and cycled using a StepOnePlus™ Real-Time PCR System. Data calculated as copies per 5 μl cDNA were converted to copies per cell based on the known input cell equivalents. Primer sequences used are listed in Supplementary Table 11.

**Inducible shRNA knockdown**. The FH1-tUTG inducible knockdown vector was digested using BsmBI (NEW ENGLAND BioLabs, #R0580S) and XhoI (NEW ENGLAND BioLabs, #R0146S) enzymes, and the annealed shRNA oligos were inserted into the digested vector using the T4 DNA ligase (ThermoFisher Scientific, #EL0014). The lentiviral particles were packaged via co-transfection of FH1-tUTG vector inserted with shRNA oligos (44 μg), pMDLg/pRRE plasmid (22 μg), pMD2.g plasmid (13.2 μg) and pRSV-Rev plasmid (11 μg) plasmids into HEK293T cells[70]. A549 or H1299 cells were transduced with the lentiviral particles in 6 cm cell culture dishes to establish inducible knockdown cell sublines. The knockdown of PLANE was induced in response to doxycycline treatment. ShRNA sequences are shown in Supplementary Table 4.

**Xenograft mouse model**. A549 cells expressing the inducible PLANE shRNAs were subcutaneously injected into the dorsal flanks of 4-week-old female nude mice (6 mice per group, Shanghai SLAC Laboratory Animal Co. Ltd., China). Tumour growth was measured every 3 days using a calliper. Mice were sacrificed after 33 days of cancer cell transplantation. Tumours were excised and measured. Studies on animals were conducted in accordance with relevant guidelines and regulations and were approved by the Animal Research Ethics Committee of the first affiliated hospital, Shanxi Medical University and Shanxi Cancer Hospital and Institute (China). All mice were housed in a temperature-controlled room (21–23 °C) with 40–60% humidity and a light/dark cycle of 12 h/12 h.

**Statistical analysis**. Statistical analysis was carried out using the GraphPad Prism 8 to assess differences between experimental groups. Statistical differences were analysed by two-tailed Student's $t$-test or one-way ANOVA test followed by Tukey's multiple comparisons. $P$ values lower than 0.05 were considered to be statistically significant.

**Reporting summary**. Further information on research design is available in the Nature Research Reporting Summary linked to this article.

## Data availability

The RNA sequencing data have been deposited in the NCBI Gene Expression Omnibus database under the accession code GSE162215 and GSE172124. The mass spectrometry proteomics data have been deposited to the ProteomeXchange Consortium (http://proteomecentral.proteomexchange.org) via the iProX partner repository with the dataset identifier PXD022747. The long noncoding RNA expression data and E2F1 mRNA expression data referenced during the study are available in a public repository from the Cancer RNA-seq Nexus dataset (http://syslab4.nchu.edu.tw/). The PLANE expression and relevant cancer patient survival data referenced during the study are available in a public repository from the GEPIA website (http://gepia.cancer-pku.cn/) under the accession codes TCGA-LUSC, TCGA-COAD, TCGA-KIRC and TCGA-UCEC. The MELTF expression and relevant cancer patient survival data were obtained from the human protein atlas website (https://www.proteinatlas.org/). The gene amplification frequency data referenced during the study are available in a public repository from the

cBioPortal website (https://www.cbioportal.org/) under the accession code TCGA PanCancer Atlas Studies. Source data are provided with this paper.

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

## Acknowledgements

This work was supported by the National Health and Medical Research Council (NHMRC; APP1147271, APP1162753, APP1177087), Cancer Council NSW Project Grant (RG20-10), Australia, and National Natural Science Foundation of China (82002442), China. The authors thank A/Professor Marco Herold (Walter and Eliza Hall Institute of Medical Research, Australia) for the plasmid FH1-tUTG, Professor Xiaoju Zhang (Respiration Department, Henan Provincial People's Hospital, Zhengzhou, China) for providing the NCI-H1975 and NCI-H226 cell lines, Professor Mian Wu (Translational Research Institute, Henan Provincial People's Hospital, Zhengzhou, China) for plasmids pcDNA3.1(+), pGL4.73[hRluc/SV40] and pSin-3 × Flag-E2F1, A/Professor Yongyan Wu (Department of Otolaryngology, the first affiliated hospital, Shanxi Medical University, Taiyuan, China) for the plasmid pEGFP-C1, respectively. The authors thank the Shanghai Luming Biological Technology Co., LTD (Shanghai, China) for providing proteomics services.

## Author contributions

X.D.Z., F.-M.S., L.J. and T. Liu designed the experiments. X.D.Z., F.-M.S and L.J. supervised the work. L.T., Y.C.F., P.L.W., T.F.Q., Y.M.Y., S.X.W., S.N.Z., C.X.T., T.La, Y.Y.Z., X.H.Z., D.Z., J.Y.W., Y.S. and H.C. performed experiments using human cell lines and tissues and related data collections; S.T.G. conducted experiments in xenograft models; S.T.G., J.N.G., L.Y.W. and X.Y.L. collected clinical samples; T.Liu, J.M.L., Y.C.F., L.J., T.La, R.F.T. and J.Y.W. carried out analysis of publicly available data and bioinformatics analysis. X.D.Z., R.F.T., F.-M.S., T.Liu and L.J. wrote the manuscript. All authors commented on the manuscript.

## Competing interests

The authors declare no competing interests.
