## [Peer Review File · Nature Communications]

REVIEWER COMMENTS

Reviewer #1 (Remarks to the Author):

Teng and colleagues present an in-depth and thorough study of a lncRNA (PLANE/MELTF-AS1) that they identified owing to its location in a region of the genome that is recurrently amplified in multiple cancers. They find through multiple orthogonal approaches that PLANE expression decreases colony-growth/tumor-like properties, that it downregulates NCOR2, and that the lncRNA is regulated by E2F. They present evidence to suggest that PLANE downregulates NCOR2 at least in part by repressing the production of a single, alternatively spliced isoform of the gene, and that this repression depends on HNRNPM, to which PLANE binds robustly. On the whole the data support the conclusions of the study, and the manuscript represents a substantial effort and body of work. Intuitively, I have some difficulty understanding how the strong effects of NCOR2 repression by PLANE can be mediated entirely by a single alternative splicing event that appears to be quite minor relative to the other isoforms of the transcript.

I have a few suggestions for the authors that would mitigate my concerns.

1) Is it possible that PLANE is regulating export of NCOR2 mRNA -- all spliced isoforms? The authors could use existing reagents to test this by fractionating RNA in WT and PLANE KD cells, and performing RT-qPCR for spliced NCOR2.

2) From my reading, the authors did not include RNA-seq methods and differential gene expression analysis methods in the methods section, preventing me from evaluating that experiment fully. I apologize if I am missing this somehow. Please also include a table that shows the expression of all genes with and without PLANE knockdown, and indicate which of those genes expression changes significantly.

3) Relatedly, was RNA-seq performed with more than one shRNA of PLANE? If not, that would be a requirement for inclusion of the RNAseq data in this study.

4) Again relatedly, the RNAseq data is the clearest data that the authors have to evaluate their alternative splicing model. Their DEseq analysis is what the entire second half of the paper is based upon. The authors should show the density of RNAseq reads over the NCOR2 gene using IGV, and use the "sashimi plot" feature of IGV to also show alternative splicing frequency within the NCOR2 gene. The authors could also show IGV evidence to support some of the additional alternative splice products they characterize in Figure 3.

5) It is difficult to evaluate the RIP data in Figure 6g and 6i showing the PLANE-dependent association between HNRNPM and NCOR2. I request that the authors use quantitative PCR to evaluate RIP signal, much in the way that qPCR was used earlier in the paper (Figure 2 for example). In this qPCR experiment, the authors should plot data in the different genotypes relative to input RNA, and should show the data for all replicates.

6) This suggestion is stylistic -- pending that the authors model holds up in subsequent experiments, in the discussion section and possibly even in the abstract, they should hedge their model a bit -- PLANE may have additional functions beyond repressing alternative splicing of NCOR2. Indeed my viewing of the data is that it suppresses protein production of NCOR2 generally, not necessarily only the alternate splice form. The paper would benefit from describing this possibility in the discussion and perhaps even making reference to it in the abstract.

Reviewer #2 (Remarks to the Author):

In the comprehensive manuscript "The pan-cancer lncRNA PLANE regulates an alternative splicing program to promote cancer pathogenesis", Teng and co-workers investigate a function of lncRNA MELTF-AS1, splicing variant 202, which the authors renamed PLANE. The authors show that the MELTF-AS1 gene is frequently amplified in cancer cells leading to overexpression of PLANE lncRNA.

In a series of experiments the authors showed that PLANE downregulation inhibits cell cycle progression and tumor growth. PLANE downregulation further deregulates alternative splicing of several genes including transcription factor NCOR2. Mechanistically, the authors provide evidence that PLANE enhances interaction of hnRNP M with NCOR2 pre-mRNA and thus regulates its splicing. The authors performed many experiments that support their conclusions. However, I have a few fundamental questions and several comments to the experimental set up and conclusions.

1. The authors claim that downregulation of PLANE enhances expression of NCOR2 protein namely via stimulation of NCOR2-202 splicing variant. However, it is unclear how the authors specifically identified the variant NCOR2-202 to be overexpressed (line 187-189 "The results showed that the NCOR2 AS variant, NCOR2-202 (ENST00000397355.1; grch37.ensembl.org), was the most highly upregulated transcript associated with knockdown of PLANE"). There is no description in Material and Methods, which platform was used for RNA-seq and how the RNA-seq was performed and analysed. Did the authors obtain long-reads or short (<150bp) reads? If the short read technology was employed how this specific splicing variant was identified? How do the authors know that the splicing variant 005 with shorter exon 45 (from variant 202) was not upregulated instead? Throughout the whole paper, there is no evidence that the whole 202 variant is produced. The authors only show changes in one particular splice site.

In addition, Western blotting shows clear induction of NCOR2 protein after PLANE knockdown (Fig. 3b). But PLANE downregulation by shRNA shows only very mild effect on exon 45 alternative splicing while production of 001 and 005 variants remains high (Fig. 3d). How do the authors explain that a mild increase of one splicing variant causes such a dramatic effect on the protein level?

2. It is not fully explained how the whole regulatory system would work. There are hundreds of PLANE lncRNA per cell while hnRNP M abundance is ~10,000x higher. Why such an abundant protein needs a low abundance navigator? The authors partially discuss this paradox in Discussion but does not provide any appropriate explanation. Most of splicing events occurs co-transcriptionally. PLANE might navigate hnRNP M locally to NCOR2 gene. If that is the case, is PLANE lncRNA located at the NCOR2 gene?

3. Line 192-193. The authors state that "Due to sequence overlaps it was not feasible to specifically confirm the increase in the NCOR2-202 AS variant using qPCR". However, the usage of downstream 5' splice site includes >130nucleotides. Why this extra sequence cannot be utilized to discriminate 202-like splicing from other splicing variants by RT-qPCR? This comment is associated with a more complex issue that involves detection of alternative splicing changes. The authors use primers that detect two products of exon 45 alternative splicing – the major longer variant using downstream 5'ss and the shorter 202-specific variant (Fig. 3d-g, Fig. 4h,j). However, PCR signal from the major longer variant looks saturated and does not likely change over a range of PCR amplification cycles and therefore cannot be properly quantified.

Other points:

- Expression of MELTF-AS transcript is complex and involves different promoters. A better graphical depiction of individual MELTF-AS1 transcripts in Fig. S1e would help to understand the Fig. S1f. Also primers designed to detect a spliced form of MELTF-AS1-202/PLANE (e.g. over exons 2 and 3 or 3 and 4) would improve detection of the spliced PLANE transcript.

- There is some blur surrounding lines in Figs. 1b and S2c. How the graphs were made? In addition, expression of MELTF, a protein coding transcript synthesized opposite to PLANE shows a similar negative survival prognosis (e.g. https://www.proteinatlas.org/ENSG00000163975-MELTF/pathology/colorectal+cancer#imid_1649104). Thus, it is unclear whether negative prognosis shown in Fig. 1b is truly associated with PLANE expression or whether it is a mere coincidence. The author should discuss this point.

- line 153 – "Furthermore, PLANE levels were correlated with E2F1 expression". Actually, R values show rather weak/no correlation.

- Fig. 2e shows reduced proliferation upon Dox induced expression of anti-PLANE shRNA in A549 and H1299 stable cell lines. What was the effect of Dox treatment on parental cell lines not harboring shRNA?

- In figure legend, there is frequently a statement that presented Data are representatives of 3 independent experiments. However, accompanying graphs lack more data points and error bars (e.g. Fig. 3d-e and f-g, Fig. 4j-k, Fig. 5c-f). The authors need to show results of all three experiments in the graph and the experimental variation.

-Line 207-209 "Moreover, knockdown of PLANE did not affect the enrichment of the transcriptional activation mark H3K4me3 and the transcriptional repression mark H3K27me3 to the PLANE promoter". Actually, ChIP actually shows decrease of H3K4me3 and increase of H3K27me3. How many times was the experiment repeated and is there any statistics available?

- Fig. S7a – exon sequences are mis-labelled. Exon 45 ends with GGG and GU marked mistakenly with blue color as part of exon 45 are actually first two nucleotides of intron removed in variants 001 and 005. And exon 46 starts with GU not with GGC.

- Fig. 7Sc – where is DFO sequence located in PLANE lncRNA? I did not find any nucleotides numbering that would identify DFO.

- Fig. 4c - I did not find a detailed description of this experiment in M&M. I assume that AS PLANE is antisense transcript to PLANE and is used as a negative control. Why then the authors do not detect it in input and IP? Without this detection, its role as negative control is questionable.

- Table S4 - The first two protein names actually point to the same protein. Correct it because it's a bit confusing.

- Fig. 5e - Does the 331-751 PLANE fragment that interacts with hnRNP M contain any hnRNP M binding site?

- Immunofluorescence staining of hnRNP M looks different in Fig. 5g and S8c, why is that? Also at Fig. 5g. SC35 and hnRNP M seems to localize next to each other rather than co-localize.

- line 310-311 "Through in silico analysis we identified a fragment enriched of consensus hnRNPM-binding sites (hnRNPM-BSS) within intron 45 of the NCOR2 pre-mRNA" – the authors identified two hnRNP M binding sites. They need to show hnRNP M binding sites throughout the whole NCOR2 transcript to substantiate the statement that hnRNP M binding sites are enriched in intron 45.

- line 351 'missing "with".'

- quantitative PCR - Normalization to an average of 3 different housekeeping genes is essential. See for example <https://www.nature.com/articles/6605851>

Reviewer #3 (Remarks to the Author):

In this manuscript, Chen F et al characterized the role of lncRNA PLANE in lung tumorigenesis found that PLANE was upregulated in cancers through copy number enhancement and E2F1-induced transcription. The authors further demonstrated that PLANE forms an RNA-RNA duplex with NCOR2 pre-mRNA at intron 45 and binds to hnRNP M to facilitate the association of hnRNP M and the intron 45, thereby suppressing NCOR2-202 splicing isoform production. lncRNA PLANE is also named as MELTF-AS1 and MF12-AS1. Many previous studies have reported that MF12-AS1 and MELTF-AS1 promoted tumorigenesis. This study termed lncRNA MELTF-AS1 or MF12-AS1 as a novel lncRNA name PLANE. PLANE also promoted tumorigenesis as previously reported.

1. PLANE was here named as pan-cancer lncRNA. However, PLANE expression was only detected in lung and colon cancers. The data about PLANE expression in other cancer types were from TCGA database. The authors also need to perform the related analyses to investigate the PLANE

expression in these cancer types from clinical samples collected by our group.

2. In supplementary Fig 1f, the expressions of these lncRNA were not detected in the same gel, indicating that their exposure time was not same. Thus, the expression levels among the five different gels were not compared.

3. The PLANE isoform reported here was different from or same as other MELTF-AS1 and MFI2-AS1 isoforms reported by previous studies. Were their functions same?

4. In supplementary Fig. 3e, the associations of PLANE and E2F1 should be analyzed in the clinical samples collected by the author's group as showed in Fig 1a.

5. In clinical samples, the relations of PLANE with NCOR2-202 should be investigated.

6. Were the functions between NCOR2-202 and classic NCOR2 same? How does NCOR2-202 promoted cell proliferation and tumorigenesis?

7. hnRNP M is as a RNA binding protein. Does hnRNP M directly bind to NCOR2 pre-mRNA independent on PLANE. Does PLANE regulate the binding of hnRNP M to NCOR2 pre-mRNA.

8. Splicing will induce an increase of a variant and a corresponding decrease of other variant. However, in this study, PLANE knockdown induced NCOR2-202 variant enhancement, but no change in other variants. Why?

9. In Fig 3b and 3c, the WB results or band models about NCOR2 were different form the RNA results.

10. NCOR2-202 was regulated by PLANE. However, NCOR2, not NCOR2-202, was further investigated together with PLANE.

11. In Fig 4h and 4i, the mutant experiments should be needed.

12. The results in Fig 4j and 4k did not indicate that splicing was associated with RNA-RNA binding.

13. In Fig 5g, Do PLANE and NCOR2 pre-mRNA co-localize with sc35?

14. In Fig 6g, the results were obvious. And a mutant, which does not bind to hnRNP M, should be need in the investigation.

15. How does the hnRNP M-binding to PLANE regulate NCOR2 pre-mRNA splicing?

Responses to Reviewers comments

Reviewer #1

We thank this reviewer's constructive comments.

1) Is it possible that PLANE is regulating export of NCOR2 mRNA -- all spliced isoforms? The authors could use existing reagents to test this by fractionating RNA in WT and PLANE KD cells, and performing RT-qPCR for spliced NCOR2.

As suggested, we tested whether PLANE regulates nuclear export of NCOR2 mRNA using subcellular fractionation in A549 cells. Using primers that detect a common region of all three major protein-coding NCOR2 alternative splice variants (NCOR2-001, NCOR2-005 and NCOR2-202) together with NCOR2-015, NCOR2-017, NCOR2-018, NCOR2-022, NCOR2-201 and NCOR2-203, we employed qPCR assays of subcellular fractions in cells with or without knockdown of PLANE. Notably, PLANE knockdown did not cause any significant changes in the proportions of these NCOR2 mRNA variants distributing to nuclear and cytoplasmic fractions, indicating that PLANE-mediated regulation of NCOR2 expression is not associated with altered nuclear export of NCOR2 mRNA variants.

These results are now shown in Supplementary Fig. 9c. Accordingly, sentences have been added in text to indicate this (lines 274-282 page 11).

2) From my reading, the authors did not include RNA-seq methods and differential gene expression analysis methods in the methods section, preventing me from evaluating that experiment fully. I apologize if I am missing this somehow. Please also include a table that shows the expression of all genes with and without PLANE knockdown, and indicate which of those genes expression changes significantly.

Apologies. We now append the full RNA-seq and differential gene expression analysis methods in the Materials and Methods section (lines 573-640, pages 21-24). In brief, for short-read RNA-seq, the Illumina HiSeq 4000 platform was used, and differential expression analysis was conducted using the DESeq2 Bioconductor package; for long-read RNA-seq, the PromethION platform was used, and differential expression analysis was carried out using the DESeq2 R package (1.6.3).

Further as requested, we have appended a table showing the expression of all genes with and without PLANE knockdown (Supplementary Data 1). Genes with significantly changed in expression are indicated. The table is referred to in text (lines 196-200, page 8).

3) Relatedly, was RNA-seq performed with more than one shRNA of PLANE? If not, that would be a requirement for inclusion of the RNaseq data in this study.

We have performed RNA-seq with two PLANE siRNAs. Among the commonly upregulated 44 transcripts, NCOR2-202 was the most highly increased one after PLANE knockdown.

These results are shown in Fig. 3a and Supplementary Data 1. Sentences have been modified/added in text and Figure legends to describe this (lines 195-200, page 8).

4) Again relatedly, the RNaseq data is the clearest data that the authors have to evaluate their alternative splicing model. Their DEseq analysis is what the entire second half of the paper is based upon. The authors should show the density of RNaseq reads over the NCOR2 gene using IGV, and use the "sashimi plot" feature of IGV to also show alternative splicing frequency within the NCOR2 gene. The authors could also show IGV evidence to support some of the additional alternative splice products they characterize in Figure 3.

We have added figures to show the density of RNA-seq reads over the NCOR2 gene using IGV together with Sashimi plots showing alternative splicing frequency within the gene (Fig. 3e & Supplementary Data 2).

As IGV evidence of the additional alternative splice products that we characterized in Figure 3 would not add significantly to the main message of the paper, we have not included the data in the manuscript. Nevertheless, an IGV-Sashimi plot of the alternative splicing frequency of SLC25A14 (as an example) is appended here for reference (Additional Fig. 1).

5) *It is difficult to evaluate the RIP data in Figure 6g and 6i showing the PLANE-dependent association between HNRNPM and NCOR2. I request that the authors use quantitative PCR to evaluate RIP signal, much in the way that qPCR was used earlier in the paper (Figure 2 for example). In this qPCR experiment, the authors should plot data in the different genotypes relative to input RNA, and should show the data for all replicates.*

As suggested, the experiments in Figure 6g and 6i were repeated using qPCR assays to quantitate the hnRNPM-BSs associated with NCOR2 pre-mRNA. New panels are now presented in Figs. 6c and 6h showing the alternative detection approach. Accordingly, the legends for these panels have been modified.

6) *This suggestion is stylistic -- pending that the authors model holds up in subsequent experiments, in the discussion section and possibly even in the abstract, they should hedge their model a bit -- PLANE may have additional functions beyond repressing alternative splicing of NCOR2. Indeed my viewing of the data is that it suppresses protein production of NCOR2 generally, not necessarily only the alternate splice form. The paper would benefit from describing this possibility in the discussion and perhaps even making reference to it in the abstract.*

We have modified/added sentences in the Abstract and Discussion to indicate that, in addition to repressing alternative splicing of NCOR2, PLANE may have other functions affecting NCOR2 expression (lines 39-40, page 2; lines 469-473, page 18).

Reviewer #2

We thank this reviewer's constructive comments.

1. *The authors claim that downregulation of PLANE enhances expression of NCOR2 protein namely via stimulation of NCOR2-202 splicing variant. However, it is unclear how the authors specifically identified the variant NCOR2-202 to be overexpressed (line 187-189 "The results showed that the NCOR2 AS variant, NCOR2-202 (ENST00000397355.1; grch37.ensembl.org), was the most highly upregulated transcript associated with knockdown of PLANE"). There is no description in Material and Methods, which platform was used for*

RNA-seq and how the RNA-seq was performed and analysed. Did the authors obtain long-reads or short (<150bp) reads? If the short read technology was employed how this specific splicing variant was identified? How do the authors know that the splicing variant 005 with shorter exon 45 (from variant 202) was not upregulated instead? Throughout the whole paper, there is no evidence that the whole 202 variant is produced. The authors only show changes in one particular splice site.

In addition, Western blotting shows clear induction of NCOR2 protein after PLANE knockdown (Fig. 3b). But PLANE downregulation by shRNA shows only very mild effect on exon 45 alternative splicing while production of 001 and 005 variants remains high (Fig. 3d). How do the authors explain that a mild increase of one splicing variant causes such a dramatic effect on the protein level?

Apologies for this oversight concerning Materials and Methods. The RNA-seq platforms and data analysis are now fully described. Briefly, for short-read RNA-seq, the Illumina HiSeq 4000 platform was used, and differential expression analysis was conducted using the DESeq2 Bioconductor package. For long-read RNA-seq, the PromethION platform was used, and differential expression analysis was carried out using the DESeq2 R package (1.6.3).

To determine alternative splicing events using short-read data, Asprofile v1.0 was employed to analyze all pairs of transcripts to determine exons included in one transcript and spliced in another from RNA-seq data (Florea L, *et al.*, *F1000Res.* 2013; 2:188). Briefly, short reads were mapped to the human genome and then assembled overlapping reads on the genome into transcript fragments using Cufflinks (Trapnell C, *et al.*, *Nat Biotechnol.* 2010; 28(5):511-515). Cufflinks represents all reads at a locus as an assembly graph, in which any two reads are connected if they overlap and have compatible splice patterns, and then traverses the graph to produce the minimum number of transcripts that can explain all of the input reads.

We initially conducted short-read RNA seq data. The results showed that “the NCOR2 AS variant, NCOR2-202 (ENST00000397355.1; grch37.ensembl.org), was the most highly upregulated transcript in cells with PLANE knockdown by siRNA1 and siRNA2” (detailed in Materials and Methods). This was followed by confirmation using long-read RAN-seq. Analysis with the DESeq2 R package similarly demonstrate that NCOR2-202 was upregulated by PLANE knockdown. Importantly, neither short-read nor long-read data showed any significant changes in the levels of NCOR2-001 and NCOR2-005. These results are shown in Figs. 3a, b & Supplementary Data 1. Sentences have been re-written/added in Results and Figure legends to describe this (lines 573-640, pages 21-24; lines 195-204, page 8).

To further confirm PLANE knockdown specifically results in an increase in the NCOR2-202 mRNA variant, we carried out serial qPCR analyses using primers recognizing various collections of NCOR2 mRNA variants. As stated in text “by use of primers recognizing the 1-138 fragment of intron 45 that is not contained by NCOR2-202, we could collectively detect the NCOR2-001/005-like mRNA variants using qPCR (Supplementary Fig. 8a). The results exhibited a decrease, albeit moderately, in cells with PLANE knocked down (Fig. 3c). In contrast, qPCR analysis using primers recognizing a common region present in NCOR2-202 and the NCOR2-001/005-like mRNA variants except for NCOR2-018 revealed an increase in expression caused by knockdown of PLANE (Fig. 3d & Supplementary Fig. 6a). Importantly, qPCR analysis using primers recognising only NCOR2-018 and NCOR2-203 displayed reduced expression in PLANE knockdown cells (Supplementary Fig. 8b)..... Taken together, these results indicate that PLANE indeed selectively inhibits the expression of the NCOR2-202 mRNA variant (Fig. 3c, d & Supplementary Figs. 6a, 8a-d).” (lines 211-229, page 9). A schematic flowchart of how we specifically identified that NCOR2-202 was upregulated in cells with PLANE knocked down has been added as a Supplementary Figure (Supplementary Fig. 8d).

We endeavored to target the 1-138 fragment present in NCOR2-001 and NCOR2-005 but not NCOR2-202 within intron 45 using siRNAs to interrogate the contribution of the increase in NCOR2-202 to the upregulation of NCOR2 protein caused by PLANE knockdown. However, we did not achieve any meaningful knockdown of

NCOR2-001 and/or NCOR2-005 using two individual siRNAs in numerous attempts (Supplementary Fig. 9b). This is now stated in Text (lines 257-262, page 10).

Nevertheless, among the three major protein-producing NCOR2 mRNA variants, NCOR2-001 and NCOR2-005 were present at relatively high levels, but NCOR2-202 levels, similar to the levels of the NCOR2 protein levels, were low in A549 and H1299 cells without knockdown of PLANE (Fig. 3d, f, g), suggesting that NCOR2 protein expression is not closely related the expression of these NCOR2-001 and NCOR2-005 mRNA variants. On the other hand, while the NCOR2-001 and NCOR2-005 levels remained unchanged, the levels of NCOR2-202 was apparently increased, concurring with the upregulation of the NCOR2 protein (Fig. 3a, b, d, f, g & Supplementary Data 1). Together, these results suggest that PLANE-mediated inhibition of NCOR2 protein expression is associated with repression of the AS event producing NCOR2-202. We have rewritten paragraphs/sentences in Results to describe and discuss this (lines 262-272, pages 10-11).

2. It is not fully explained how the whole regulatory system would work. There are hundreds of PLANE lncRNA per cell while hnRNP M abundance is ~10,000x higher. Why such an abundant protein needs a low abundance navigator? The authors partially discuss this paradox in Discussion but does not provide any appropriate explanation. Most of splicing events occurs co-transcriptionally. PLANE might navigate hnRNP M locally to NCOR2 gene. If that is the case, is PLANE lncRNA located at the NCOR2 gene?

We thank the reviewer for pointing out this important point and the opportunity to improve our Discussion. Naturally, hnRNP family members make multi-faceted contributions to nucleic acid metabolism and hnRNPM is not only involved in the regulation of NCOR2 alternative splicing. Thus, the abundance of hnRNPM reflects its vastly broader regulatory roles in cells compared to PLANE. The role of PLANE is therefore conceivably related to providing specificity to hnRNPM's interaction with NCOR2 pre-mRNA. However, our current evidence suggests PLANE does not have a direct role in promoting association between hnRNPM and DNA since the NCOR2 gene was not captured with PLANE using nuclear extracts from A549 and H1299 cells. We have added sentences in Discussion to better discuss the possibilities of how the PLANE-hnRNPM might function.

The additional results are now shown in Fig. 4e and described in text (lines 401-406, page 15).

3. Line 192-193. The authors state that “Due to sequence overlaps it was not feasible to specifically confirm the increase in the NCOR2-202 AS variant using qPCR”. However, the usage of downstream 5' splice site includes >130nucleotides. Why this extra sequence cannot be utilized to discriminate 202-like splicing from other splicing variants by RT-qPCR? This comment is associated with a more complex issue that involves detection of alternative splicing changes. The authors us primers that detect two products of exon 45 alternative splicing – the major longer variant using downstream 5'ss and the shorter 202-specific variant (Fig. 3d-g, Fig. 4h,j). However, PCR signal from the major longer variant looks saturated and does not likely change over a range of PCR amplification cycles and therefore cannot be properly quantified.

As correctly pointed out by the Reviewer, it is feasible to use primers recognizing the 138 nt fragment that is lacking in NCOR-202 to differentiate NCOR2-202-like from other NCOR2 mRNA variants. The use of such primers collectively detects NCOR2-001 and NCOR2-005 as well as NCOR2-002, NCOR2-018, NCOR2-201 and NCOR2-203. Indeed, qPCR results showed a decrease, albeit moderately, in cells with PLANE knockdown. In contrast, qPCR analysis using primers recognizing a common region present in NCOR2-202 and all these NCOR2 mRNA variants (except for NCOR2-018) revealed an increase in expression caused by knockdown of NCOR2. Of note, qPCR analysis using primers recognizing only NCOR2-018 and NCOR2-203 displayed reduced expression in cells with PLANE knockdown. Collectively, these results indicate that PLANE selectively inhibits NCOR2-202 mRNA variant expression. These results are now shown in Fig. 3c, d and Supplementary Fig. 8a, b, d and described in Text (lines 219-229, page 9).

We have replaced the figures shown in Figs. 3d (now 3f), 3f (now 3h), 4h, and 4j using RT-PCR results from experiments with varying amplification cycle numbers (30 vs 32 cycles) to avoid saturation of any product band. Accordingly, semi-quantitation of the bands has been re-performed and corresponding results are now shown in Figs. 3f, 3h, and 4k.

- *Expression of MELTF-AS transcript is complex and involves different promoters. A better graphical depiction of individual MELTF-AS1 transcripts in Fig. S1e would help to understand the Fig. S1f. Also primers designed to detect a spliced form of MELTF-AS1-202/PLANE (e.g. over exons 2 and 3 or 3 and 4) would improve detection of the spliced PLANE transcript.*

We now present better graphical depictions of the MELTF-AS1 transcripts in Supplementary Fig. 1e. We also carried out RT-PCR analysis using primers spanning across exons 2 and 3. The results similarly showed the longest isoform of MELTF-AS1 was markedly more abundant than others in multiple cancer cell lines. These results are now shown in Supplementary Fig. 1f. Sentences have been added/modified in text to indicate this (lines 107-112, page 5).

- *There is some blur surrounding lines in Figs. 1b and S2c. How the graphs were made? In addition, expression of MELTF, a protein coding transcript synthesized opposite to PLANE shows a similar negative survival prognosis (e.g. https://www.proteinatlas.org/ENSG00000163975-MELTF/pathology/colorectal+cancer#imid_1649104). Thus, it is unclear whether negative prognosis shown in Fig. 1b is truly associated with PLANE expression or whether it is a mere coincidence. The author should discuss this point.*

The blur surrounding lines in Fig. 1b and S2d (now Supplementary Fig. 2d) appear to be caused by conversion to the pdf format since these artefacts are not evident in the original Powerpoint (.ppt) files. We used a different method to convert the figures and no blur is evident.

It is true that there is a negative association between high MELTF mRNA expression and patient survival in multiple types of cancers, including colorectal cancer. However, our results indicate that there is no regulatory interaction between PLANE and the MELTF gene (Supplementary Fig. 4b-d). Sentences have been added in Results describe and discuss this (lines 162-169, page 7).

- *line 153 – “Furthermore, PLANE levels were correlated with E2F1 expression”. Actually, R values show rather weak/no correlation.*

We have modified the sentence which now reads “there was a trend that high PLANE levels were associated with E2F1 expression levels in diverse cancer types (Supplementary Fig. 3e)”. Moreover, we conducted analysis using freshly isolated esophageal squamous cell carcinoma (ESCC) and lung adenocarcinoma (LUAD) tissues. qPCR assessment showed that the levels of PLANE were positively associated with E2F1 mRNA levels. These results are now presented in Supplementary Fig. 3f and described in Results (lines 155-158, page 7).

- *Fig. 2e shows reduced proliferation upon Dox induced expression of anti-PLANE shRNA in A549 and H1299 stable cell lines. What was the effect of Dox treatment on parental cell lines not harboring shRNA?*

At the levels used for the inducible system, Dox treatment does not affect cell proliferation in the corresponding parental cell lines. We have added additional data in Supplementary Fig. 5d to indicate this.

- *In figure legend, there is frequently a statement that presented Data are representatives of 3 independent experiments. However, accompanying graphs lack more data points and error bars (e.g. Fig. 3d-e and f-g, Fig. 4j-k, Fig. 5c-f). The authors need to show results of all three experiments in the graph and the experimental variation.*

We have added more data points and error bars to Figs. 3g, 3i, 4k, 6e and 6g. Legends to these figure panels have been accordingly modified to indicate this.

-Line 207-209 “Moreover, knockdown of PLANE did not affect the enrichment of the transcriptional activation mark H3K4me3 and the transcriptional repression mark H3K27me3 to the PLANE promoter”. Actually, ChIP actually shows decrease of H3K4me3 and increase of H3K27me3. How many times was the experiment repeated and is there any statistics available?

These experiments were repeated three times and we now append a statistical analysis based on all experiments (new panels have been added to Supplementary Fig. 7c). Consistent with our original conclusion there were no statistically significant differences in the amounts of H3K4me3 and H3K27me3 associated with the *NCOR2* promoter, respectively, before and after knockdown of PLANE. We also substituted the gel images with alternate ones to ensure reader comprehension of the images fully matches this conclusion.

- Fig. S7a – exon sequences are mis-labelled. Exon 45 ends with GGG and GU marked mistakenly with blue color as part of exon 45 are actually first two nucleotides of intron removed in variants 001 and 005. And exon 46 starts with GU not with GGC.

These errors have been corrected in Fig. S7a (now Supplementary Fig. 8a).

- Fig. 7Sc – where is DFO sequence located in PLANE lncRNA? I did not find any nucleotides numbering that would identify DFO.

We modified this figure panel to now clearly indicate the location and sequence of the DFO (Supplementary Fig. 11a).

- Fig. 4c - I did not find a detailed description of this experiment in M&M. I assume that AS PLANE is antisense transcript to PLANE and is used as a negative control. Why then the authors do not detect it in input and IP? Without this detection, its role as negative control is questionable.

AS PLANE represents the antisense PLANE transcript used as the negative control. The methods related to the Fig. 4c experiment (i.e. domain-specific chromatin isolation by RNA purification or dChIRP) have been described in the Materials and Methods (lines 757-758, page 27). There was no detectable band for AS PLANE with the ‘sense’ primers used for the three different ‘sense’ transcripts used. To assuage this concern, we added reactions to detecting the antisense transcript for both input and dChIRP panels for this figure.

- Table S4 - The first two protein names actually point to the same protein. Correct it because it's a bit confusing.

This error in Table S4 (now Supplementary Table 5) has been corrected. The second (same name) protein has been deleted from the table.

- Fig. 5e - Does the 331-751 PLANE fragment that interacts with hnRNP M contain any hnRNP M binding site?

The 331-751 PLANE fragment that interacts with hnRNPM contains an hnRNPM binding site. This is now illustrated in Supplementary Fig. 12a.

- Immunofluorescence staining of hnRNP M looks different in Fig. 5g and S8c, why is that? Also at Fig. 5g. SC35 and hnRNP M seems to localize next to each other rather than co-localize.

The images shown in Fig. 5g have been replaced with clearer examples showing co-localization of SC35 and hnRNPM and also better consistency with the images in Supplementary Fig. 8c (now Supplementary Fig. 12c).

- line 310-311 “Through in silico analysis we identified a fragment enriched of consensus hnRNPM-binding sites (hnRNPM-BSs) within intron 45 of the NCOR2 pre-mRNA” – the authors identified two hnRNP M binding sites. They need to show hnRNP M binding sites throughout the whole NCOR2 transcript to substantiated the statement that hnRNP M binding sites are enriched in intron 45.

The hnRNP M binding sites occurring throughout the whole NCOR2 transcript are now shown in Supplementary Data 3. Indeed, the fragment within intron 45 of the NCOR2 pre-mRNA is one of multiple fragments across the NCOR2 transcript that are enriched of hnRNPM-binding sites (hnRNPM-BSs). Examination of a random selection of 3 additional such fragments (105067-105217, 158355-158512 and 176814-177141) showed that 158355-158512 and 176814-177141 were co-precipitated with hnRNPM, whereas 105067-105217 did not bind to the protein. Nevertheless, in contrast to the binding of the fragment within intron 45 to hnRNPM, the associations between 158355-158512 as well as 176814-177141 and hnRNPM were not affected by PLANE knockdown, suggesting that PLANE-mediated facilitation of binding of hnRNPM within the fragment at intron 45 of the NCOR2 pre-mRNA is specific. These results are shown in Supplementary Fig. 12g, h. Sentences have been modified/added in text to indicate this (lines 374-386, pages 14-15).

- line 351 ‘missing “with”’.

This error has been corrected (line 430, page 16).

- quantitative PCR - Normalization to an average of 3 different housekeeping genes is essential. See for example <https://www.nature.com/articles/6605851>

We re-carried out all qPCR experiments and normalized the results against an average of three housekeeping genes (GAPDH, β -actin and 18s rRNA) (Figs. 1h, 2a, d, 3c, d, Supplementary Figs. 2e, 3d, f, 4c, d, 5b, f, 7a, b, 8b, c, 9a, b, d, 11b). Sentences have been added in Materials and Methods to describe this modified approach (lines 650-652, page 24) including citation of the reference provided by the Reviewer (lines 1071-1072, page 35). The findings remain ostensibly unchanged.

Reviewer #3

We thank this reviewer’s constructive comments.

1. PLANE was here named as pan-cancer lncRNA. However, PLANE expression was only detected in lung and colon cancers. The data about PLANE expression in other cancer types were from TCGA database. The authors also need to perform the related analyses to investigate the PLANE expression in these cancer types from clinical samples collected by our group.

We have extended our analysis of PLANE expression to additional cancer types, including freshly isolated esophageal squamous cell carcinoma (ESCC) and breast cancer (BRCA) samples in comparison with adjacent normal tissues. The results showed that PLANE was indeed increased in expression in esophageal squamous carcinoma and breast cancer. These results are now shown in Supplementary Fig. 2a. Sentences have been modified/added in text to describe this (lines 119-121, page 5).

2. In supplementary Fig 1f, the expressions of these lncRNA were not detected in the same gel, indicating that their exposure time was not same. Thus, the expression levels among the five different gels were not compared.

We re-performed the experiment shown in Supplementary Fig 1f with all samples now resolved in a single gel (same exposure). The findings were ostensibly the same, but we agree this approach adds further confidence. Furthermore, the whole gel image is appended here for appraisal (Additional Fig. 2). The legend to this figure was modified to indicate that the images were all derived from a single gel.

3. *The PLANE isoform reported here was different from or same as other MELTF-AS1 and MFI2-AS1 isoforms reported by previous studies. Were their functions same?*

As stated in the text, “..... the longest isoform (of MELTF-AS1/MFI2-AS1) was markedly more abundant than others in multiple cancer cell lines, including A549 and H1299 LUAD, NCI-H226 lung squamous cell carcinoma (LUSC), HCT116 colon adenocarcinoma (COAD), MCF-7 BRCA and Eca109 esophageal squamous cell carcinoma (ESCC). Previously reported studies involving MELTF-AS1/MFI2-AS1 did not specify which isoform(s) were investigated but we anticipate they would likely have examined the most abundant isoform which we now name PLANE (lines 107-112, page 5). Furthermore, in agreement with previous reports showing that MELTF-AS1/MFI2-AS1 plays an oncogenic role in various cancer types (Wei Y, *et al.*, *Biomed Pharmacother.* 2020; 125:109890; Li C, *et al.*, *Cell Prolif.* 2019; 52(4):e12632), our functional studies demonstrated that PLANE promotes cancer cell proliferation. We have added sentences in the Discussion to elaborate this point for the reader (lines 444-447, page 17) and appropriate references have now been cited (lines 1033-1038, page 35).

4. *In supplementary Fig. 3e, the associations of PLANE and E2F1 should be analyzed in the clinical samples collected by the author’s group as showed in Fig 1a.*

As requested, we analyzed the relationship between PLANE and E2F1 mRNA expression in freshly isolated esophageal squamous cell carcinoma (ESCC) and lung adenocarcinoma (LUAD) using qPCR. The results showed that the levels of PLANE were positively associated with E2F1 mRNA levels, thus independently supporting findings obtained with the TCGA data. These results are now shown in Supplementary Fig. 3f and described in Results (lines 156-158, page 7).

5. *In clinical samples, the relations of PLANE with NCOR2-202 should be investigated.*

We have analyzed the relationship between PLANE and NCOR2-202 expression levels in freshly isolated esophageal squamous cell carcinoma (ESCC) and lung adenocarcinoma (LUAD) using RT-PCR. Semi-

quantitative results showed that there was no significant relationship between PLANE and NCOR2-202 mRNA levels. These results are now shown in Supplementary Fig. 8e and described in Results (lines 230-233, page 9).

6. *Were the functions between NCOR2-202 and classic NCOR2 same? How does NCOR2-202 promoted cell proliferation and tumorigenesis?*

As indicated in the text, NCOR2-202 is one of the NCOR2 pre-mRNA alternative splicing variants (lines 196-200, page 8). As stated in Introduction and Discussion, NCOR2 as a transcriptional corepressor functions by way of a platform that links chromatin modifying enzymes such as HDACs and transcription factors to regulate transactivation of downstream genes involved in many cellular processes including cell survival and proliferation (lines 59-71, page 3; lines 426-429, page 16). Its role in cell proliferation and tumorigenesis varies widely in cell type- and tissue-type-dependent manner. For example, a number of studies have demonstrated a tumour suppressive role of NCOR2 in cancers such as LUAD, head and neck squamous cell carcinoma, non-Hodgkin lymphoma and osteosarcoma (Ghoshal P, *et al.*, *Cancer Res.* 2009; 69(10):4380-4387; Song L, *et al.*, *Cancer Res.* 2005; 65(11):4554-4561; Compton LA, *et al.*, *Cancer Cell.* 2010; 17(4):315-316; Alam H, *et al.*, *Cancer Res.* 2018; 78(14):38343848), whereas it has also been shown to promote cell survival and proliferation in some other cancer types such as breast cancer (Gong C, *et al.*, *Clin Cancer Res.* 2018; 24(15):3681-3691) (lines 431-437, pages 16-17).

Here we demonstrated a tumour suppressive function of the NCOR2-202 isoform. However, whether each of the other NCOR2 isoforms has any specific effect on cancer pathogenesis remains to be defined (lines 437-441, page 17). We also discuss the possibility that the NCOR2 pre-mRNA splicing pattern may change in a context-dependent fashion (Goodson M, *et al.*, *J Bio Chem.* 2011; 286(52):44988-44999) (lines 442-444, page 17).

7. *hnRNP M is as a RNA binding protein. Does hnRNP M directly bind to NCOR2 pre-mRNA independent on PLANE. Does PLANE regulate the binding of hnRNP M to NCOR2 pre-mRNA.*

The results shown in Fig. 6c indicate that hnRNPM can bind to the hnRNPM-binding sites (hnRNPM-BSs) within intron 45 of the NCOR2 pre-mRNA in cells with PLANE knocked down, although the amounts of hnRNPM associating with the hnRNPM-BSs were reduced, suggesting that PLANE plays a role in facilitating the binding between hnRNPM and NCOR2 pre-mRNA within intron 45. These results are shown in Fig. 6c and described in text (lines 374-382, pages 14-15).

8. *Splicing will induce an increase of a variant and a corresponding decrease of other variant. However, in this study, PLANE knockdown induced NCOR2-202 variant enhancement, but no change in other variants. Why?*

We have added new figure panels in Supplementary Fig. 8b showing that the increases in NCOR2-202 isoform expression associated with PLANE knockdown correspond with decreases in other NCOR2 mRNA variants. Sentences have been added in the text to indicate this (lines 224-226, page 9).

9. *In Fig 3b and 3c, the WB results or band models about NCOR2 were different form the RNA results.*

The Western blotting data for NCOR2 are different between Fig. 3b and 3c (now Fig. 3d and Supplementary Fig. 9a) because they involve PLANE knockdown and overexpression, respectively. In Fig. 3b (now Fig. 3d), PLANE knockdown causes increase in the expression of NCOR2 transcripts (primers recognizing NCOR2-001, NCOR2-202 and NCOR2-005) while the Western blot images show corresponding upregulation of NCOR2 protein (the efficiency of PLANE knockdown is shown in Fig. 2d). In comparison, Fig. 3c (now Supplementary Fig. 9a) shows that overexpression of PLANE causes downregulation of NCOR2 protein levels.

We reviewed the figure legends and text associated with these figures to ensure the meaning was clear to the readers.

10. *NCOR2-202 was regulated by PLANE. However, NCOR2, not NCOR2-202, was further investigated together with PLANE.*

As indicated in the text, our investigation was indeed focused on NCOR2-202, specifically the biological impact of PLANE regulation of AS production of NCOR2-202. “To investigate the biological impact of PLANE regulation of AS production of NCOR2-202, we tested the effect of siRNAs targeting common regions of NCOR2-001, NCOR2-005 and NCOR2-202 on inhibition of cell proliferation caused by PLANE knockdown (Supplementary Fig. 6a), which conceivably reflected the consequence of inhibition NCOR2-202 expression, as NCOR2 protein upregulation in PLANE knockdown cells was primarily caused by the increase in NCOR2-202 (Fig. 3a-g & Supplementary Fig. 8d). For simplicity, we hereafter refer to these siRNAs as NCOR2 siRNAs” (lines 284-289, page 11).

11. *In Fig 4h and 4i, the mutant experiments should be needed.*

Experiments shown in Fig 4h and 4i were indeed carried out in cells transfected with plasmids carrying mutant PLANE with the DFO deleted. The mutant form of PLANE was detected using RT-PCR as shown in Fig. 4c. This is a widely accepted approach in similar investigations (Hosono Y, *et al.*, *Cell*. 2017; 171(7):1559-1572.e20; Munschauer M, *et al.*, *Nature*. 2018; 561(7721):132-136; Shi Q, *et al.*, *Nat Commun*. 2020; 11(1):5513; Wang Z, *et al.*, *Cancer Cell*. 2018; 33(4):706-720.e9) The experiments leading up to these figures were conducted with the non-mutated forms of PLANE (e.g. Fig. 4c).

12. *The results in Fig 4j and 4k did not indicate that splicing was associated with RNA-RNA binding.*

We agree, these experiments alone do not show that the splicing is associated with RNA-RNA binding events. However, as indicated in the text, the conclusion that ‘the formation of the RNA-RNA duplex is required for the PLANE effects on NCOR2 pre-mRNA AS’ was built upon the collective data of this and the preceding experiments. We had stated this in the text, i.e. ‘taken together with preceding data (Fig. 4c-k & Supplementary Fig. 11b).

13. *In Fig 5g, Do PLANE and NCOR2 pre-mRNA co-localize with sc35?*

We have carried out fluorescence in situ hybridization (FISH) experiments. The results showed that a proportion of PLANE was colocalized with SC35, whereas little NCOR2 pre-mRNA was found to colocalize with SC35. These results are shown in Supplementary Fig. 12d, e. Sentences have been added in text to indicate this (lines 366-369, page 14).

14. *In Fig 6g, the results were obvious. And a mutant, which does not bind to hnRNP M, should be need in the investigation.*

Beyond knockdown of PLANE, we have confirmed that PLANE, but not the mutant PLANE (with the 331-751 fragment deleted) that cannot bind to hnRNPM, facilitated the binding between hnRNPM and the pre-mRNA (Fig. 6h). Moreover, the results obtained after PLANE knockdown are further reinforced by the rescue experiments performed with an shRNA-resistant PLANE (Fig. 6c).

These results have been shown in Fig. 6c, h and described in text and Figure Legends (lines 391-399, page 15).

15. *How does the hnRNP M-binding to PLANE regulate NCOR2 pre-mRNA splicing?*

As per our Discussion, hnRNP family proteins are broadly involved in regulating RNA maturation. In particular, hnRNPM influences AS in a variety of pre-mRNAs of cancer-associated genes (lines 475-479, page 18). Our

results demonstrate that PLANE facilitates the binding of hnRNPM to the hnRNPM binding sites (hnRNPM-BSs) within intron 45 of the NCOR2 pre-mRNA. This binding leads to suppression of the NCOR2-202-generating AS event (lines 480-495, page 18). Nevertheless, while we do not know the exact mechanisms involved, our work establishes a broader precedent for alternative splicing that is likely shared in other genes. It would be highly interesting to further investigate the detailed mechanisms, but it is beyond the scope of this manuscript.

REVIEWERS' COMMENTS

Reviewer #1 (Remarks to the Author):

I thank the reviewers for addressing all of my comments.

Reviewer #2 (Remarks to the Author):

I am satisfied with the changes made in the revised version of the manuscript NCOMMS-20-47452A "The pan-cancer lncRNA PLANE regulates an alternative splicing program to promote cancer pathogenesis". I have just two minor comments:

- NCOR2 gene produces many various mRNA isoforms and new variants are mentioned in the new text at p. 9 (NCOR2-017, 018, 203). In general, it would significantly help to understand the whole logic of NCOR2-202 detection (depicted at Supplementary fig. 8d) if all NCOR2 mRNA variants mentioned in the text are included in Supplementary fig. 6a, including primers used for their detection.

- p. 11, l. 274 a typo in "PLANE".

Reviewer #3 (Remarks to the Author):

The authors have responded well our concerns.

Responses to Reviewers comments

Reviewer #2

We thank this reviewer's further constructive comments.

- NCOR2 gene produces many various mRNA isoforms and new variants are mentioned in the new text at p. 9 (NCOR2-017, 018, 203). In general, it would significantly help to understand the whole logic of NCOR2-202 detection (depicted at Supplementary fig. 8d) if all NCOR2 mRNA variants mentioned in the text are included in Supplementary fig. 6a, including primers used for their detection.

As requested, we now illustrate the additional NCOR2 mRNA variants (NCOR2-017, 018, 203) within Supplementary Fig 6a. The legends for this panel have been modified accordingly.

- p. 11, l. 274 a typo in "PLANE".

This error has been corrected.

Yours Sincerely,

Xu Dong Zhang